# DISCOVERING GROUP STRUCTURES VIA UNITARY REPRESENTATION LEARNING

**Dongsung Huh**
IBM Research
`huh@ibm.com`

## ABSTRACT

Discovering group structures within data poses a fundamental challenge across diverse scientific domains. A key obstacle is the non-differentiable nature of group axioms, hindering their integration into deep learning frameworks. To address this, we introduce a novel differentiable approach leveraging the representation theory of finite groups. Our method features a unique network architecture that models interactions between group elements via matrix multiplication of their representations, along with a regularizer promoting the unitarity of these representations. The interplay between the network architecture and the unitarity condition implicitly encourages the emergence of valid group structures. Evaluations demonstrate our method's ability to accurately recover group operations and their unitary representations from partial observations, achieving significant improvements in sample efficiency and a $\times 1000$ speedup over the state of the art. This work lays the foundation for a promising new paradigm in automated algebraic structure discovery, with potential applications across various domains, including automatic symmetry discovery for geometric deep learning.

## 1 INTRODUCTION

Identifying algebraic structures, particularly groups, has been a cornerstone of progress across diverse scientific fields. In mathematics, groups formalize the concepts of symmetry and transformation, underpinning abstract algebra, geometry, topology, and number theory. In physics, group theory is indispensable for understanding the fundamental laws of nature, from classifying elementary particles to formulating quantum field theory. Within computer science, groups play key roles in cryptography, coding theory, and algorithm design. Furthermore, in deep learning, group theory informs the design of symmetry-aware architectures (*e.g.*, convolutional and equivariant neural networks), enhancing parameter efficiency, generalization, and enabling applications on non-Euclidean domains (Bronstein et al., 2021).

Despite their profound significance, uncovering group structures within data remains a significant challenge, traditionally requiring substantial domain expertise and human intuition. A key obstacle is the inherent non-differentiability of the defining criteria for groups — the group axioms — which impedes their direct integration into gradient-based learning frameworks. In this work, we introduce a novel method for automatically discovering groups and their unitary representations, leveraging the representation theory of finite groups. Our approach uses a novel network architecture that models interactions between group elements through the product of their matrix representations, along with a regularizer promoting their unitarity. This interplay naturally encourages the emergence of valid group structures, without direct evaluation of the non-differentiable group axioms.

This work demonstrates the effective embedding of fundamental group structure criteria within the differentiable learning framework, paving the way for automated discovery of algebraic structures.

## 2 GROUPS AND REPRESENTATIONS

Algebraic structures — sets endowed with operations satisfying specific axioms — offer a powerful framework for studying abstract mathematical objects and their interactions. Among these structures, groups are foundational building blocks of abstract algebra, underpinning the construction of

more complex entities like rings and fields. Furthermore, the well-established theory of group representations provides a crucial tool for analyzing group structures by mapping their abstract properties to the concrete domain of linear algebra. In this section, we present a concise overview of groups and their representations, emphasizing key concepts pertinent to our work.

**Groups**   A group $(G, \circ)$ is a set $G$ with a binary operation $\circ$ that satisfies four axioms: Closure: $\forall a, b \in G, \, a \circ b \in G$. Associativity: $(a \circ b) \circ c = a \circ (b \circ c)$. Identity: There exists an identity element $e \in G$ such that for all $g \in G$, $g \circ e = e \circ g = g$. Inverse: For every $g \in G$, there exists a unique inverse element $g^{-1}$ such that $g \circ g^{-1} = g^{-1} \circ g = e$.

**Representations**   A representation of a group $(G, \circ)$ on a vector space $V$ is a *group homomorphism* $\varrho \colon G \to \mathrm{GL}\,(V)$ that preserves the group structure: *i.e.*

$$\varrho(g_1 \circ g_2) = \varrho(g_1)\varrho(g_2), \qquad \forall g_1, g_2 \in G. \tag{1}$$

In essence, it maps each group element to an invertible linear transformation on the vector space, ensuring that the composition of transformations mirrors the group operation. For a finite-dimensional vector space of dimension $n$, we can choose a basis to identify $GL(V)$ as $GL(n, K)$, the group of $n \times n$ invertible matrices over the field $K$.

**Unitary Representations**   A representation $\varrho$ of a group $(G, \circ)$ is called *unitary* if for every $g \in G$, $\varrho(g)$ is a unitary transformation, *i.e.* preserves the inner product. This property makes unitary representations particularly well-behaved and amenable to analysis. Notably, the *Unitarity Theorem* guarantees that for many important classes of groups, such as compact and finite groups, every finite-dimensional representation is equivalent to a unitary one. Unitary representations naturally arise in the study of quantum systems, and have deep connections to other areas of mathematics, e.g., harmonic analysis and operator algebras.

**Irreducible Representations**   A representation is considered *reducible* if it can be decomposed into a direct sum of smaller representations via a similarity transform, leading to a block-diagonal matrix form where each block corresponds to a simpler representation. *Irreducible* representations (irreps), on the other hand, cannot be further decomposed and serve as the fundamental building blocks for constructing all possible group representations.

**Regular Representations**   Every group $(G, \circ)$ possesses an inherent action on itself that can be viewed as a permutation, where each group element rearranges the other elements. The *regular* representation uses the permutation's basis vectors to construct a linear representation. It is decomposible into a direct sum of the *complete* set of irreps, where each irrep appears with a multiplicity equal to its dimension. Moreover, its trace, also known as *character*, is a simple function:

$$\mathrm{Tr}[\varrho(g)] = n \text{ if } g = e, \; 0 \text{ otherwise.} \tag{2}$$

**Real vs Complex Representations**   Complex representations $(K = \mathbb{C})$ provide a rich mathematical framework for analyzing group structures in representation theory. We utilize this framework to establish the theoretical foundations of our approach in Sections 4 and 5. However, for finite groups, real representations $(K = \mathbb{R})$ often suffice in practice,[1] offering advantages in implementation and visualization. Our empirical results in Sections 6 and 7 thus utilize real representations.

## 3   BACKGROUND

**Binary Operation Completion**   In this study, we adopt the Binary Operation Completion (BOC) problem (Power et al., 2022) as our experimental setting. BOC entails completing the Cayley table of a binary operation over a finite set of abstract symbols. This problem isolates the fundamental challenge of discovering group structures solely from interactions between elements, eliminating the confounding influence of extraneous factors. Consequently, BOC provides a crucial theoretical framework for analyzing structure learning within the discrete symbolic domain, serving a role analogous to that of matrix completion in the continuous domain.

---

[1]For finite groups, every complex representation can be realized over the real numbers with a doubling of the dimension.

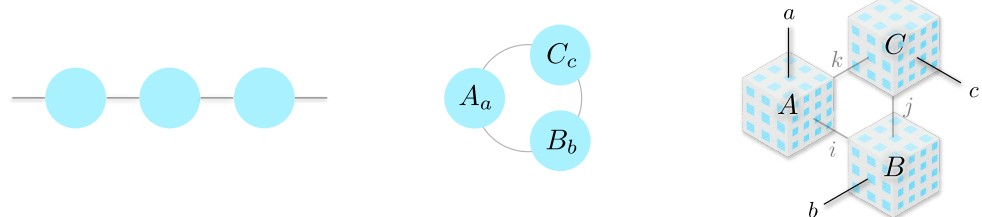

Figure 1: Illustration of matrix and tensor products. Nodes are factors and edges are indices. (Left) Matrix product. (Middle) Matrix product with trace operation. (Right) HyperCube product.

**Matrix Completion**   Matrix completion aims to infer missing entries of a partially observed matrix, assuming an underlying low-rank structure. Classical methods typically enforce this assumption by either imposing explicit rank constraints (Burer and Monteiro, 2003) or by minimizing the nuclear norm as a convex proxy for rank (Fazel et al., 2001; Candès and Recht, 2009; Recht et al., 2010; Candes and Tao, 2010). Matrix completion provides a theoretical foundation for numerous applications, spanning recommender systems, data imputation, compressed sensing.

**Implicit Complexity Metric**   Recent research has revealed that deep matrix factorization networks, when regularized with $L_2$ regularization (or initialized with small weights), implicitly define a complexity metric that approximates rank, such as the nuclear norm or Schatten norm (Srebro et al., 2004; Gunasekar et al., 2017). Furthermore, these implicit approaches have demonstrated superior performance in matrix completion, especially with limited data (Arora et al., 2019).

**Our Contributions: Bridging the Gap**   While Binary Operation Completion (BOC) shares conceptual similarities with matrix completion, its discrete symbolic nature poses distinct challenges. To bridge this gap, we reformulate the BOC problem into a tensor completion problem. We then propose a novel solution rooted in the representation theory of finite groups: a specialized tensor-factorization architecture coupled with a novel regularizer. This approach implicitly defines a complexity metric that acts as a differentiable proxy for the group criterion, thus providing a learning-based methodology for discovering group structures directly from data.

## 4   MODELING FRAMEWORK

**Notations and Definitions**   We use the following capital symbols for order-3 tensor factors: $A, B, C$. $A_a$ denotes the matrix slice of $A$ at the first index $a$ and $A_a^\dagger$ denotes its conjugate transpose. $A_a B_b$ denotes the matrix product of $A_a$ and $B_b$. Einstein convention is used throughout, where a repeated index implies contraction: *e.g.*, $A_a A_a^\dagger \equiv \sum_a A_a A_a^\dagger$, unless otherwise specified.

### 4.1   LINEARIZED FRAMEWORK: BINARY OPERATIONS AS BILINEAR MAPS

Consider a binary operation $\circ : S \times S \to S$ over a finite set $S$ containing $n$ elements: *i.e.* $a \circ b = c$, where $a, b, c \in S$. To facilitate modeling, we linearize the problem by considering a homomorphism $\phi : (S, \circ) \to (V, \mathcal{D})$, where $V$ is a vector space and $\mathcal{D} : V \times V \to V$ is a bilinear map over $V$, such that $\mathcal{D}(\phi(a), \phi(b)) = \phi(a \circ b)$. Concretely, by choosing the vector space $V = \mathbb{C}^n$ with a basis (for instance, encoding each element as a one-hot vector), the bilinear map $\mathcal{D}$ can be represented by an order-3 tensor $D \in \mathbb{C}^{n \times n \times n}$, whose entries are

$$D_{abc} = 1 \text{ if } a \circ b = c, \ 0 \text{ otherwise.} \tag{3}$$

where the elements of $S$ are used as tensor indices for clarity. Hereafter, we will use $D$ to denote the ground-truth data tensor to be learned by the model.

The linearized framework reveals that any binary operation over a finite set can be fully modeled by a bilinear map, or equivalently, by its tensor representation. Crucially, this framework transforms BOC into a tensor completion problem, where we recover the missing entries of $D$ from the observed entries in the training set.

## 4.2 HyperCube Parameterization

To solve the tensor completion problem, we train a model tensor $T$ to recover the data tensor $D$. However, treating the entries of $T$ as independent model parameters would prevent the model from leveraging the structural relationships between observed and unobserved entries, limiting its ability to generalize.

To address this, we introduce HyperCube factorization (Fig. 1), a structured parameterization that factorizes the model tensor as a product of three order-3 factors (*i.e. cubes*) $A, B, C \in \mathbb{C}^{n \times n \times n}$:

$$T_{abc} = \frac{1}{n} \operatorname{Tr}[A_a B_b C_c] = \frac{1}{n} \sum_{ijk} A_{aki} B_{bij} C_{cjk}. \tag{4}$$

This architecture employs matrix embeddings to represent the elements of the set $S$. Factors $A$ and $B$ serve as embedding dictionaries, mapping each element $a$ and $b$ to their respective matrix embeddings: $A_a$ and $B_b$.[2] The interaction between $a$ and $b$ is then modeled as the matrix multiplication: $A_a B_b$. Factor $C$ then acts as an *unembedding* dictionary, mapping the result back to the space of $S$.

This architecture is inspired by the representation theory of finite groups, which also employs matrix multiplication to model group operations (eq (1)). A key advantage of this approach is that it directly encodes the associativity axiom of groups through the inherent associativity of matrix multiplication, providing a strong inductive bias for capturing group structures. In contrast, conventional tensor factorization methods typically use lower-order factors to reduce model complexity but lack this useful inductive bias. As a result, they are less effective at modeling group structures (see Appendix D for a detailed comparison).

## 4.3 HyperCube Regularizer

The model is trained by minimizing the following regularized objective:

$$\mathcal{L} = \mathcal{L}_o(T; D) + \epsilon \mathcal{H}(A, B, C), \tag{5}$$

where $\mathcal{L}_o$ is a differentiable loss on the model tensor $T$ (*e.g.*, squared error over the training data) and $\mathcal{H}$ is the *HyperCube regularizer*, which penalizes the Jacobian of $T$ with respect to the parameters:

$$\mathcal{H} \equiv \left\| \frac{\partial T}{\partial A} \right\|_F^2 + \left\| \frac{\partial T}{\partial B} \right\|_F^2 + \left\| \frac{\partial T}{\partial C} \right\|_F^2 = \frac{1}{n} \operatorname{Tr} \left[ A_a^\dagger A_a B_b B_b^\dagger + B_b^\dagger B_b C_c C_c^\dagger + C_c^\dagger C_c A_a A_a^\dagger \right], \tag{6}$$

which can be viewed as a *dual* to standard $L_2$ regularization: $\|A\|_F^2 + \|B\|_F^2 + \|C\|_F^2$. In subsequent sections, we demonstrate that eq (6) encourages the factors to learn full-rank, unitary matrix embeddings, contrasting with $L_2$ regularization, which promotes low-rank solutions.

The *Unitarity Theorem* of representation theory guarantees that for compact and finite groups, every finite-dimensional representation is equivalent to a unitary representation. Therefore, the unitarity bias of the HyperCube regularizer can leverage this theorem to promote solutions within the space of unitary matrix embeddings without loss of generality. This focused learning within a relevant subspace of representations leads to faster convergence and improved sample complexity.

## 4.4 Internal Symmetry of Model

The over-parameterized eq (4) implies the presence of internal symmetries that leave the model unchanged. For instance, one can introduce arbitrary invertible matrices $M_I, M_J, M_K$ and their inverses between the factors as $\tilde{A}_a = M_K^{-1} A_a M_I$, $\tilde{B}_b = M_I^{-1} B_b M_J$, and $\tilde{C}_c = M_J^{-1} C_c M_K$. These yield an equivalent parameterization of $T$, since $\operatorname{Tr}[\tilde{A}_a \tilde{B}_b \tilde{C}_c] = \operatorname{Tr}[A_a B_b C_c]$. These symmetry transformations can be understood as changing the internal basis coordinate to represent the factors.

Note that while the model loss $\mathcal{L}_o(T)$ is invariant under such coordinate changes, the regularizer $\mathcal{H}(A, B, C)$ is not. However, the regularizer is invariant under *unitary* basis changes, in which the introduced matrices are unitary, such that $MM^\dagger = M^\dagger M = I$. Therefore, the regularizer imposes a stricter form of symmetry. This leads to the following Proposition.

**Proposition 4.1.** *If $A, B, C$ form an optimal solution of the regularized loss eq (5), then any unitary basis changes leave the solution optimal, but non-unitary basis changes generally increase the loss.*

---

[2]This embedding process is closely related to the generalized Fourier transform on groups (See Appendix I).

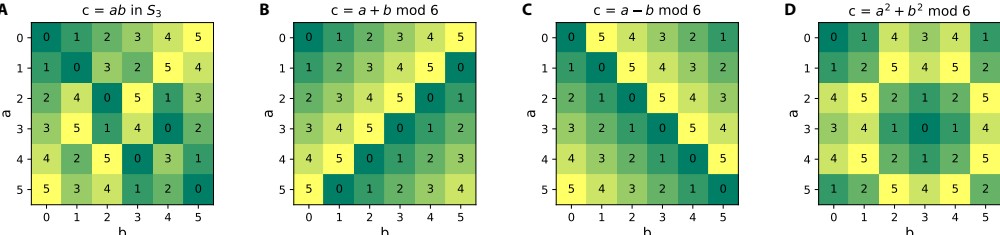

Figure 2: "Multiplication" tables (*i.e.* Cayley tables) of small binary operations: symmetric group $S_3$, modular addition, subtraction, and squared addition. Elements of $S_3$ are illustrated in Figure 8.

## 5 ANALYZING HYPERCUBE'S INDUCTIVE BIAS

While HyperCube eq (4) does not explicitly restrict the model's hypothesis space, the regularizer eq (6) induces a strong implicit bias on the model. In this section, we introduce key concepts for analyzing this inductive bias. See Appendix G for proofs.

**Lemma 5.1** (Balanced Condition). *At stationary points of eq* (5)*, imbalance terms vanish to zero:*

$$\xi_I = \xi_J = \xi_K = 0, \tag{7}$$

*where* $\xi_I = A_a^\dagger(C_c^\dagger C_c)A_a - B_b(C_c C_c^\dagger)B_b^\dagger$, $\xi_J = B_b^\dagger(A_a^\dagger A_a)B_b - C_c(A_a A_a^\dagger)C_c^\dagger$, *and* $\xi_K = C_c^\dagger(B_b^\dagger B_b)C_c - A_a(B_b B_b^\dagger)A_a^\dagger$ *are the imbalances across edge* $i, j$*, and* $k$*, respectively.*

The following statements demonstrate that the regularizer promotes a unitarity condition.

**Definition 5.2** (Contracted Unitarity). A factor $A$ is *C-unitary* if it satisfies the following: $A_a A_a^\dagger$, $A_a^\dagger A_a \propto I$ (*with* contracting the repeated index $a$).

**Proposition 5.3.** *C-unitary factors satisfy the balanced condition eq* (7)*, given that they share a common scalar multiple of the identity matrix: i.e.*

$$A_a A_a^\dagger = A_a^\dagger A_a = B_b B_b^\dagger = B_b^\dagger B_b = C_c C_c^\dagger = C_c^\dagger C_c \equiv n\alpha^2 I, \tag{8}$$

**Lemma 5.4.** *Under the fixed Frobenius norm, all C-unitary factors are stationary points of the regularizer* $\mathcal{H}$.

Remarkably, we also observe a stronger form of unitarity in the converged solutions.

**Definition 5.5** (Slice Unitarity). A factor $A$ is *S-unitary* if every matrix slice of $A$ is a scalar multiple of an unitary matrix: *i.e.* $A_a A_a^\dagger = A_a^\dagger A_a \equiv \alpha_{A_a}^2 I$ (*without* contracting the repeated index $a$).

**Observation 5.6.** *When optimizing the regularized loss eq* (5)*, C-unitary solutions are consistently achieved via S-unitarity, in which eq* (8) *reduces to* $\sum_a \alpha_{A_a}^2 = \sum_b \alpha_{B_b}^2 = \sum_c \alpha_{C_c}^2 = n\alpha^2$.

Although the exact mechanism driving S-unitarity remains an open question, this observation emphasizes the strong unitarity bias induced by the HyperCube regularizer. In Section 6, we rigorously demonstrate the learning dynamics, revealing a consistent reduction in imbalance and unitarity measures during optimization with HyperCube regularization. These findings compellingly suggest that unitarity is not merely a byproduct, but a fundamental attribute of HyperCube's optimal solutions.

## 6 ANALYSIS ON SMALL-SCALE BOC EXPERIMENTS

We begin with a detailed qualitative analysis of how HyperCube learns small-scale binary operations. First, we examine the model's learning dynamics on the group operation in $S_3$ (Section 6.1), followed by analysis of the learned matrix embeddings (Section 6.2). Finally, we extend these results to other operations from Figure 2 (Section 6.3).

### 6.1 LEARNING DYNAMICS ON SYMMETRIC GROUP $S_3$

Figure 3 compares the effect of different regularization strategies on the model's learning dynamics on the symmetric group $S_3$ with 60% of the Cayley table sampled as training data. (See also

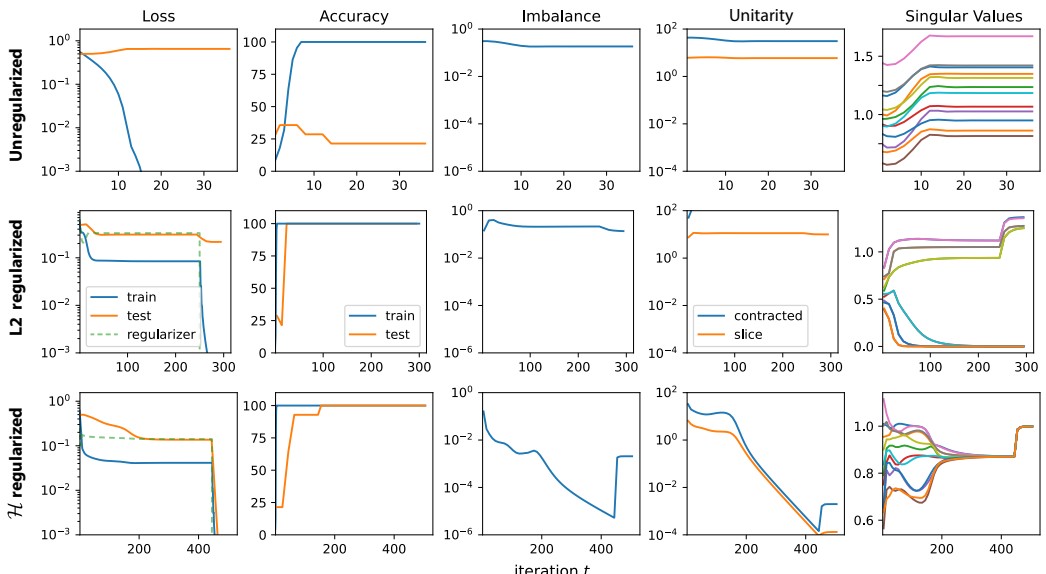

Figure 3: Optimization trajectories on the $S_3$ dataset with 60% training data fraction. (Top) Unregularized, (Middle) $L_2$-regularized, and (Bottom) $\mathcal{H}$-regularized training. Column 3 shows the average imbalance $(\|\xi_I\|_F^2 + \|\xi_J\|_F^2 + \|\xi_K\|_F^2)^{1/2}$, and column 4 shows deviation from C-unitarity $\|\sum_a A_a A_a^\dagger / n - \alpha^2 I\|_F^2$ and S-unitarity $\|A_a A_a^\dagger - \alpha_{A_a}^2 I\|_F^2$, averaged over all factors and slices. Column 5 shows normalized singular values of unfolded factors $A, B, C$.

Figure 15 for a direct visualization of the evolution of the model tensor and parameters.) Similar analysis on the learning of the modular addition operation is presented in Figure 18 and 19.

In the absence of regularization, the model quickly memorizes the training dataset, achieving perfect training accuracy, but fails to generalize to the test dataset. Also, the singular values of the unfolded factors remain largely unchanged during training, indicating minimal internal structural changes.

Under $\mathcal{H}$ regularization, the model also first memorizes the training data, but then continues to improve on the test set. A critical turning point is observed around $t \approx 200$, marked by a sudden collapse of the singular values towards a common value, signifying convergence to a unitary solution. Simultaneously, the C/S-unitarity and imbalance measures rapidly decrease to zero. This internal restructuring coincides with a substantial improvement in test performance, achieving 100% test accuracy, highlighting its crucial role in enabling generalization. Notably, when the regularization coefficient $\epsilon$ drops to 0 around $t = 450$, both the train and test losses plummet to 0, confirming perfect recovery of $D$.

In contrast, under $L_2$ regularization, the model converges to a low-rank solution, as evidenced by a portion of the singular values decaying to zero. Although it manages to achieve perfect test accuracy in this specific case,[3] $L_2$ regularization fails to reduce test loss to zero, indicating imperfect recovery of $D$. Figure 4 further confirms these findings, demonstrating that only $\mathcal{H}$-regularization accurately recovers the group operation, while the $L_2$-regularized solution significantly deviates from $D$. This result underscores the importance of learning full-rank, unitary solutions in recovering group operations.

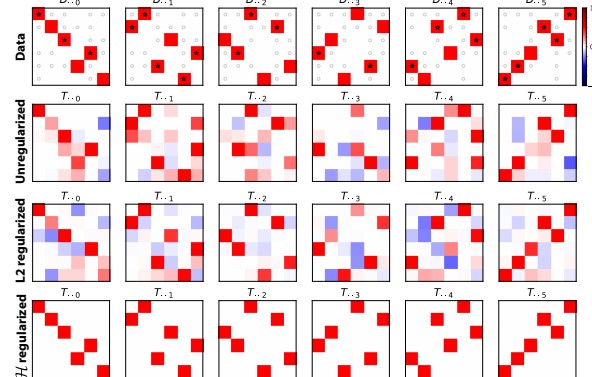

Figure 4: Model tensor $T$ trained on the $S_3$ dataset. Training data are marked by stars (1s) and circles (0s).

---

[3]This is not always the case. For example, $L_2$ regularization only achieves $\sim$60% test accuracy in the modular addition task. See Figure 18.

## 6.2 HYPERCUBE LEARNS UNITARY GROUP REPRESENTATIONS

We analyze the structure of the learned factors by applying the unitary basis change described in Section 4.4. The results are visualized in Figure 16. While the raw factor values may appear unstructured (top panel), a simple basis change reveals remarkable properties (middle panel).

First, the learned factors share a common embedding: $A_g = B_g = C_g^\dagger$, for all $g \in G$. Furthermore, as illustrated in Figure 17, multiplication of the factor slices respects the underlying group operation: $A_{g_1} A_{g_2} = A_{g_1 \circ g_2}$, demonstrating the group homomorphism property from eq (1). The slices are also verifiably unitary matrices. These properties collectively imply that the learned factors form a unitary matrix representation $\varrho$ of the group:

$$A_g = B_g = C_g^\dagger = \varrho(g). \tag{9}$$

Further structures are revealed in a block-diagonalizing basis (bottom panel of Figure 16), where the factors show the complete set of irreducible representations (irreps) contained in the regular representation of the group, including the trivial (1-dim), sign (1-dim), and duplicate standard representations (2-dim). The trace of the factor slices also satisfies eq (2), confirming that $\varrho$ is indeed a regular representation of the group. Notably, this representation is unique up to similarity transformations.

**Key Operating Mechanism** These results reveal the operating mechanism of HyperCube on groups. Applying eq (9) and the homomorphism property of $\varrho$, the model eq (4) can be expressed as

$$T_{abc} = \frac{1}{n} \operatorname{Tr}[\varrho(a)\varrho(b)\varrho(c)^\dagger] = \frac{1}{n} \operatorname{Tr}[\varrho(a \circ b \circ c^{-1})], \tag{10}$$

where the unitarity of $\varrho$ is used for the unembedding map: $C_g = \varrho(g)^\dagger = \varrho(g^{-1})$. Applying eq (2) for regular representations yields the exact reconstruction of the data tensor, $T_{abc} = D_{abc}$, since $a \circ b \circ c^{-1} = e$ is equivalent to $a \circ b = c$ in eq (3). Notably, this mechanism universally applies for all finite groups. This insight leads to the following conjecture:

**Conjecture 6.1.** *Subject to the constraint $T = D$, where $D$ represents a group operation table, the unitary group representation eq* (9) *is the unique minimizer of HyperCube Regularizer eq* (6) *up to unitary basis changes, and the minimum value is $\mathcal{H}^*(D) = 3\|D\|_F^2 = 3n^2$.*

**Shared-Embedding** Eq (9) reveals that, for group operations, the same embedding is used across all symbol positions. This motivates tying the embeddings across factors, resulting in a parameter-efficient model tailored for learning group operations: HyperCube-SE (shared embedding).

## 6.3 DISCOVERING UNITARY REPRESENTATIONS BEYOND TRUE GROUPS

We extend our analysis to HyperCube trained on the remaining small operation tasks from Figure 2. In each case, the model accurately recovers the underlying operations from a small subset (60%) of training examples. Remarkably, the model learns closely related representations across these tasks (Figure 20), even though the operations deviate from strict group axioms.

**Modular Addition** ($a + b = c$) forms the cyclic group $C_6$. As expected, HyperCube learns the regular representation $\varrho(g)$ of $C_6$ in its factors, as described by eq (9).

**Modular Subtraction** ($a - b = c$) violates associativity and therfore is not a true group. Surprisingly, HyperCube still learns the same representation as addition but with transposed factors: $A_g^\dagger = B_g = C_g = \varrho(g)$. This reflects the equivalence: $a - b = c \Leftrightarrow a = b + c$.

**Modular Squared Addition** ($a^2 + b^2 = c$) violates the inverse axiom. Still, HyperCube learns the same representation as addition for elements with unique inverses (e.g., $0, 3$). For others, it learns *duplicate* representations reflecting the periodicity of squaring modulo: *e.g.*, $A_2 = A_4$ since $2^2 = 4^2 (\mathrm{mod}\ 6)$.

These results highlight the remarkable flexibility of HyperCube's inductive bias: Even for *group-like* operations (i.e., those deviating from strict group axioms), HyperCube often discovers meaningful unitary representations and recovers the full Cayley table. This finding highlights the potential of unitary representations as a powerful tool for understanding binary operations in broader contexts than group theory.

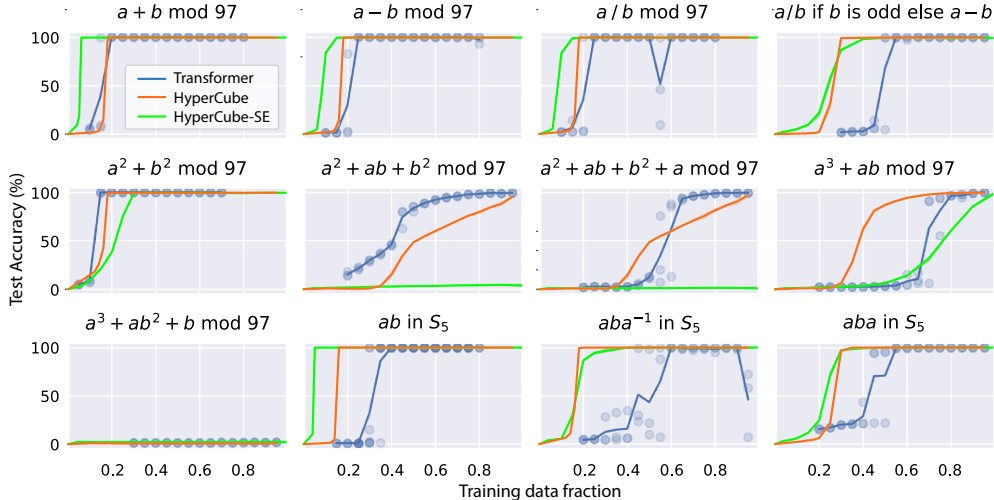

Figure 5: Generalization performance (test accuracy) of HyperCube and HyperCube-SE shown as functions of training data fraction across a diverse set of BOC tasks. The results of the Transformer baseline from Power et al. (2022) are also shown for comparison.

## 7 RESULTS ON DIVERSE BOC TASKS

We evaluate HyperCube and HyperCube-SE on diverse BOC datasets from Power et al. (2022), encompassing a wide spectrum of group and non-group operations (details in Appendix B). These problems are significantly larger than our previous examples, with dimensions ranging from $n = 97$ to 120. Figure 5 plots the test accuracy of models as functions of training data fraction over various BOC tasks.

### 7.1 HYPERCUBE PRIORITIZES GROUPS OVER NON-GROUP OPERATIONS

HyperCube exhibits a clear preference for learning operations that admit unitary representations. For these "simple" tasks, including group ($a+b$ and $S_5$) and *group-like* operations ($a-b$, $a/b$ and $a^2+b^2$), HyperCube demonstrates remarkable generalization, achieving perfect test accuracy with $\sim$18% of the data. This strong generalization extends even to the group conjugation operation ($aba^{-1}$ in $S_5$), despite its lack of unitary representations. Conversely, for more "complex" operations, such as $aba$ in $S_5$, conditional operations, and high-order polynomials, HyperCube necessitates significantly more data for effective generalization.

HyperCube-SE exhibits similar behavior, but with an even stronger emphasis on prioritizing group structures. It further differentiates between group and *group-like* operations, requiring even less data ($\sim$5%) for group operations to attain perfect test accuracy on group operations.

For *group-like* operations, HyperCube-SE remains competitive with HyperCube in terms of test accuracy, despite its inherent limitation in recovering unitary representations due to the shared-embedding constraint (Eq. (9)). Moreover, HyperCube-SE shows a further reduction in generalization performance compared to HyperCube on "complex" operations like high-order polynomials, consistent with its heightened prioritization of group structures.

### 7.2 HYPERCUBE'S IMPLICIT COMPLEXITY METRIC

In the previous section, we categorized tasks as "simple" or "complex" without a rigorous definition. To address this, we now leverage the intrinsic complexity metric implicitly defined by HyperCube. We formally define the complexity of an operation as the minimum regularizer loss, denoted $\mathcal{H}^*$, attained when fitting the fully observed Cayley table $D$ (*i.e.*, under the constraint $T = D$).

This metric closely aligns with the intuitive notion of complexity (Figure 6). Group operations achieve the minimum complexity of $\mathcal{H}^* = 3\|D\|_F^2$, signifying their inherent simplicity within the

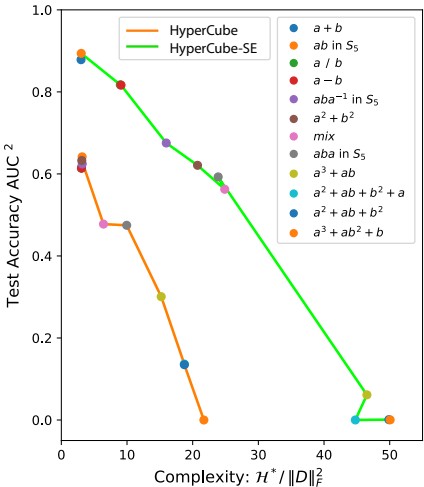

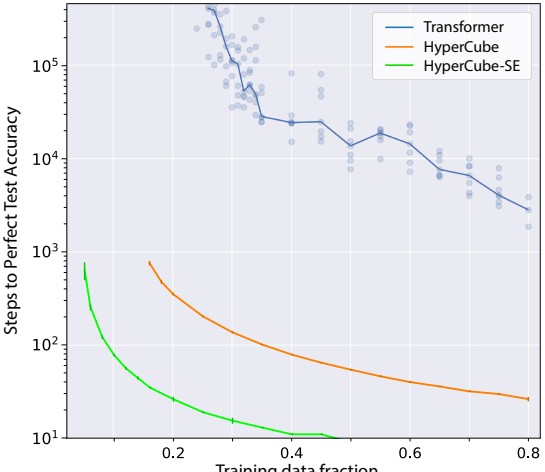

Figure 6: Complexity vs Generalizability (AUC ≡ Area Under Curve).

Figure 7: Number of training steps to achieve perfect test accuracy on the $S_5$ task.

HyperCube framework. In contrast, more "complex" tasks, such as high-order polynomials, incur substantially higher complexity costs.

Figure 6 illustrates the generalization trend as a function of complexity, revealing a clear monotonic relationship: as task complexity increases, generalizability (measured by the total area under the test accuracy curve in Figure 5) decreases. This observation parallels the well-known relationship in matrix completion, where higher matrix rank (analogous to higher complexity) generally leads to poorer generalization and requires more data for effective learning. This underscores the critical role of our proposed complexity metric in determining the generalization bound for BOC.

## 7.3 COMPARISON TO TRANSFORMER

To rigorously evaluate HyperCube, we compare against a Transformer baseline from Power et al. (2022). Crucially, this Transformer baseline was meticulously tuned for peak performance due to its inherent sensitivity to hyperparameters. [4] In contrast, HyperCube demonstrates notable robustness across a broad range of hyperparameter configurations (see Appendix C).

**Test Accuracy Performance**    Figure 5 illustrates that Transformer accuracy trends mirror Hyper-Cube's, requiring more data for increasingly "complex" tasks. However, Transformers favor commutative operations (e.g., $a+b$, $a^2+ab+b^2$) over non-commutative ones (e.g., $a-b$, $S_5$ tasks). This likely stems from Transformers' shared vector embeddings across all input locations (Power et al., 2022; Liu et al., 2022), which misaligns their inductive bias with group structure learning. Overall, HyperCube achieves comparable or slightly better generalization than Transformers in test accuracy across most tasks.

**Learning Speed**    HyperCube drastically outperforms Transformers in learning speed (Figure 7). As reported by Power et al. (2022), Transformer's learning progress on BOC is remarkably slow, requiring orders of magnitude more time to generalize to the test set than to fit the training set. This phenomenon, known as "grokking," becomes exacerbated with less training data. This has been widely observed in subsequent works across various deep learning architectures (Nanda et al., 2022; Liu et al., 2022; Chughtai et al., 2023).

In contrast, HyperCube exhibits exceptional learning speed, converging to perfect test accuracy 100 times faster than the Transformer baseline in most cases, while also requiring less data. HyperCube-SE, which employs shared-embedding of symbols similarly to Transformers, achieves an additional

---

[4]These include learning rate, batch size, weight decay, dropout, update noise level, and optimizer type. See Section 3.3 and A.1.2 of Power et al. (2022) for further details.

$10\times$ speedup and requires only $5\%$ of the data for perfect generalization. This dramatic $1000\times$ improvement in learning speed demonstrates the effectiveness of HyperCube's inductive bias toward group structures.

## 8 CONCLUSION

This work introduced a novel differentiable framework for discovering the structure of finite groups and their representations. Through theoretical analysis and empirical validation, we demonstrated that our proposed model exhibits a strong inductive bias towards learning group structures and their unitary representations. Furthermore, we have elucidated an implicit complexity metric inherent in our model, which quantifies the model's prioritization for discovering group structures and provides insights into the generalization properties in recovering binary operations. Crucially, this inductive bias is universal, directed towards the general algebraic structure of all groups, rather than being specific to any particular group or symmetry.

This research pioneers new opportunities for employing deep learning to automatically uncover group structures from data — a challenge with far-reaching implications across diverse scientific domains. Prominent potential applications include: **Automatic Symmetry Discovery**: Identifying symmetries in complex systems, such as physical systems or molecular structures. **Representation Learning**: Learning meaningful representations of data that capture underlying algebraic relationships. **Algorithmic Reasoning**: Developing deep learning models capable of symbolic reasoning and algorithmic problem-solving.

**Related Works**  Prior works on group discovery, particularly symmetry-focused approaches, are related to our work. However, existing symmetry-based methods typically require external information specifying the symmetry structure. For instance, Anselmi et al. (2019) leverage data augmentation with known orbit information, while Forestano et al. (2023) similarly presume an oracle providing orbit information. Yang et al. (2023) employs a semi-supervised approach to infer orbit information, and Zhou et al. (2021) utilizes meta-learning, assuming a shared group convolution structure across tasks. In contrast, our approach uniquely derives a universal inductive bias towards the general algebraic structure of groups, independent of specific symmetries. Consequently, our method is complementary to, and potentially synergistic with, these existing symmetry-focused techniques.

**Scalability**  HyperCube's tensor-factorization architecture can lead to substantial memory and computational costs, scaling as $O(n^3)$. However, efficient parallelization of `einsum` operations enables near-constant runtime complexity on GPUs (Appendix F). Moreover, we demonstrate that constraining embeddings to band-diagonal matrices can effectively reduce memory and computational costs to $O(n^2)$ (Appendix E), underscoring the potential for scaling HyperCube to larger problem sizes.

**Limitations**  Our analysis is primarily focused on Binary Operation Completion (BOC) tasks and currently necessitates prior knowledge of the group size, $n$. Furthermore, while our method is not directly applicable to continuous Lie groups in its present form, we anticipate that an analogous differentiable approach can be developed to encode the axioms of Lie algebras.

**Open Questions**  This work naturally leads to several open questions for future research. Prominent among these are deriving rigorous generalization bounds for BOC tasks and formally proving the optimality of unitary representations, as suggested by Observation 5.6 and Conjecture 6.1. Furthermore, extending our methodology to accommodate multiple symbols and operations beyond binary operations would significantly broaden its scope and applicability.

### ACKNOWLEDGMENTS

The authors thank Nima Dehmamy and Ken Clarkson for their helpful discussions, which greatly benefited this work.

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

## A  TRAINING PROCEDURE

The factor tensors are initialized with entries randomly drawn from a normal distribution: $\mathcal{N}(0, 1/\sqrt{n})$. We employ full-batch gradient descent to optimize the regularized loss with learning rate of $0.5$ and momentum of $0.5$. For the small scale experiments in Section 6, the HyperCube regularizer coefficient is set to $\epsilon = 0.1$. For the larger scale experiments in Section 7, we use $\epsilon = 0.05$ for HyperCube and $\epsilon = 0.01$ for HyperCube-SE. See Appendix C for a discussion of hyperparameter sensitivity. Each experiment quickly runs within a few minutes on a single GPU.

**$\epsilon$-scheduler**  To overcome the limitations in standard regularized optimization, which often prevents full convergence to the ground truth ($D$), we employ $\epsilon$-scheduler: Once the model demonstrates sufficient convergence (*e.g.*, the average imbalance falls below a threshold of $10^{-5}$), the scheduler sets the regularization coefficient $\epsilon$ to $0$. This allows the model to fully fit the training data. The effect of $\epsilon$-scheduler on convergence is discussed in Appendix G.3.

The main implementation of HyperCube is shown below.

```python
import torch

def HyperCube_product(A,B,C):
    return torch.einsum('aij,bjk,cki->abc', A,B,C) / A.shape[0]

def HyperCube_regularizer(A,B,C):
    def helper(M,N):
      MM = torch.einsum('aim,aij->mj', M,M)
      NN = torch.einsum('bjk,bmk->jm', N,N)
      return torch.einsum('mj,jm->', MM, NN)
    return (helper(A,B) + helper(B,C) + helper(C,A) ) / A.shape[0]
```

## B  LIST OF BINARY OPERATIONS

Here is the list of binary operations from Power et al. (2022) that are used in Section 7 (with $p = 97$).

- (add) $a \circ b = a + b \pmod{p}$ for $0 \le a, b < p$. (Cyclic Group)
- (sub) $a \circ b = a - b \pmod{p}$ for $0 \le a, b < p$.
- (div) $a \circ b = a/b \pmod{p}$ for $0 \le a < p, 0 < b < p$.
- (cond) $a \circ b = [a/b \pmod{p} \text{ if } b \text{ is odd, otherwise } a - b \pmod{p}]$ for $0 \le a, b < p$.
- (quad1) $a \circ b = a^2 + b^2 \pmod{p}$ for $0 \le a, b < p$.
- (quad2) $a \circ b = a^2 + ab + b^2 \pmod{p}$ for $0 \le a, b < p$.
- (quad3) $a \circ b = a^2 + ab + b^2 + a \pmod{p}$ for $0 \le a, b < p$.
- (cube1) $a \circ b = a^3 + ab \pmod{p}$ for $0 \le a, b < p$.
- (cube2) $a \circ b = a^3 + ab^2 + b \pmod{p}$ for $0 \le a, b < p$.
- ($ab$ in $S_5$) $a \circ b = a \cdot b$ for $a, b \in S_5$. (Symmetric Group)
- ($aba^{-1}$ in $S_5$) $a \circ b = a \cdot b \cdot a^{-1}$ for $a, b \in S_5$.
- ($aba$ in $S_5$) $a \circ b = a \cdot b \cdot a$ for $a, b \in S_5$.

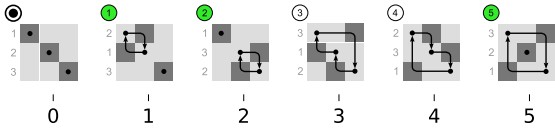

Figure 8: Elements of the symmetric group $S_3$ illustrated as permutations of 3 items. Green color indicates *odd* permutations, and white indicates *even* permutations. Adapted from https://en.wikipedia.org/wiki/Symmetric_group.

## C HYPERPARAMETER SENSITIVITY ANALYSIS

We tested HyperCube across a wide range of hyperparameter settings, including learning rate, regularization coefficient, and weight initialization scale. Figure 9 shows the final test accuracy and Figure 10 shows the number of training steps to achieve 100% test accuracy across a subset of tasks from Appendix B under a fixed training budget of 1000 training steps.

HyperCube exhibits robust performance over the range of hyperparameter settings. Notably, increasing the learning rate or regularization coefficient primarily raises the convergence speed without significantly affecting the final test accuracy. The learning dynamics starts to become unstable at large learning rate (lr = 1.5) or regularization coefficient ($\epsilon = 0.1$). The weight initialization scale has no effect on either the final test accuracy or the convergence speed.

This robustness, particularly to weight initialization scale and regularization strength, is noteworthy. Deep neural networks exhibit a saddle point with zero Hessian at zero weights (Kawaguchi, 2016) which becomes a local minimum under $L_2$ regularization. This local minimum can cause the network weights to collapse to zero when initialized with small values or under strong regularization. (This mechanism also promotes low-rank solutions in $L_2$-regularized deep neural networks.)

In contrast, HyperCube's quartic regularization loss, also featuring zero Hessian at zero weights, maintains the saddle point at zero. The absence of local minimum at zero prevents weight collapse, contributing to significantly robust learning dynamics and promoting the emergence of full-rank unitary representations in HyperCube.

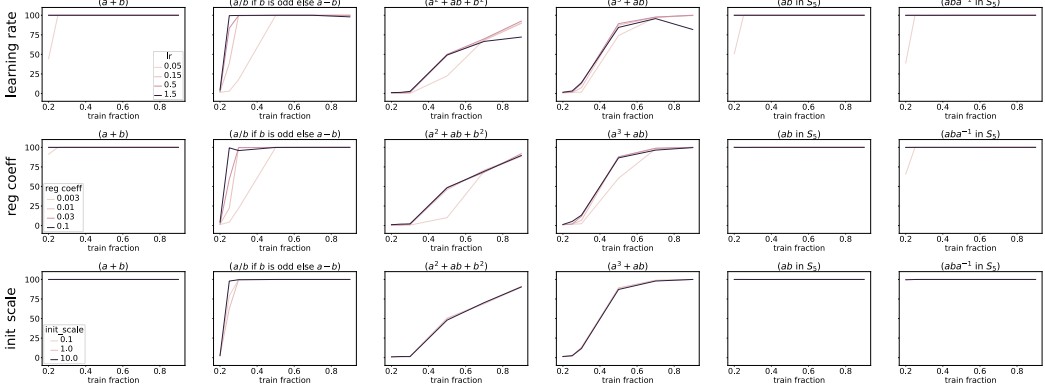

Figure 9: **Test accuracy vs Hyperparameters** : (Top) learning rate, (Middle) regularization strength, and (Bottom) weight initialization scale. Trained under a fixed training budget of 1000 steps. Default hyperparameter setting: lr = 0.5, reg coeff $\epsilon = 0.05$, init scale = 1.0.

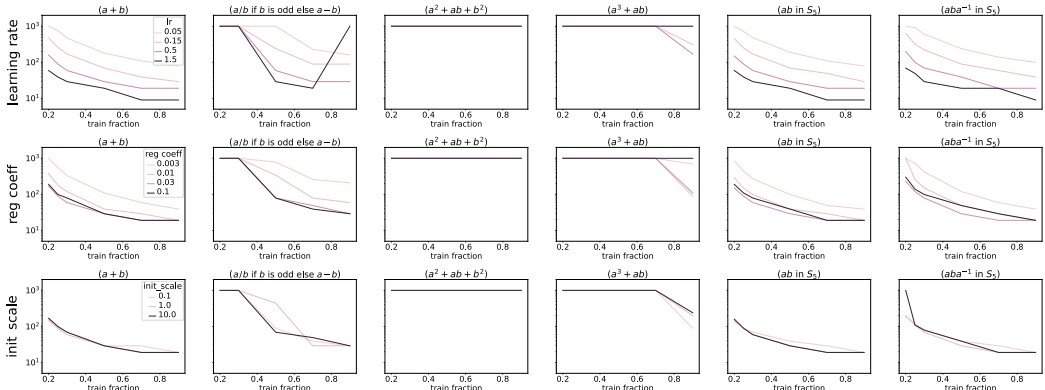

Figure 10: **Steps to 100% accuracy vs Hyperparameters** : Same settings as Fig 9, but showing the number of training steps to achieve 100% test accuracy.

# D  ALTERNATIVE TENSOR FACTORIZATIONS

HyperCube distinguishes itself from conventional tensor factorization architectures, which typically employ lower-order, matrix factors for decomposition: *e.g.*, Tucker and CP decomposition. This difference is crucial for capturing the rich structure of binary operations.

**Tucker Decomposition** (Tucker, 1966) employs a core tensor $M$ and three matrix factors:

$$T_{abc} = \frac{1}{n} \sum_{i,j,k} M_{ijk} A_{ai} B_{bj} C_{ck}, \tag{11}$$

While flexible, Tucker decomposition suffers from a critical limitation: In eq (11), the role of matrix factors is limited to simply mapping individual *external* indices to individual *internal* indices (e.g. $A$ maps $a$ to $i$). This presents a recursive challenge, since learning the algebraic relationships between the external indices $(a, b, c)$ in $T$ requires learning the relationships between the internal indices$(i, j, k)$ in $M$, which is not inherently simplifying the core learning problem. Consequently, Tucker decomposition severely overfits the training data and fails to generalize to unseen examples (Figure 11).

**CP Decomposition**  CP decomposition utilizes only matrix factors for decomposition:

$$T_{abc} = \frac{1}{n} \sum_{k} A_{ak} B_{bk} C_{ck}. \tag{12}$$

This is equivalent to[5] HyperCube with diagonal embeddings (*i.e.* $A_{aki} = A_{ak}\delta_{ki}$, $B_{bij} = B_{bi}\delta_{ij}$, $C_{cjk} = C_{cj}\delta_{jk}$), since

$$\sum_{ijk} A_{aki} B_{bij} C_{cjk} = \sum_{ijk} A_{ak} B_{bi} C_{cj} \delta_{ki} \delta_{ij} \delta_{jk} = \sum_{k} A_{ak} B_{bk} C_{ck}. \tag{13}$$

Therefore, while CP decomposition can fully capture commutative Abelian groups (e.g modular addition), which admit diagonal representations (*i.e.*, $1 \times 1$ irreps) in $K = \mathbb{C}$, it lacks the expressive power to capture more complex operations. In experiments (Figure 11), CP decomposition indeed shows reasonable performance only for the modular addition task, struggling to generalize to other structures in data.

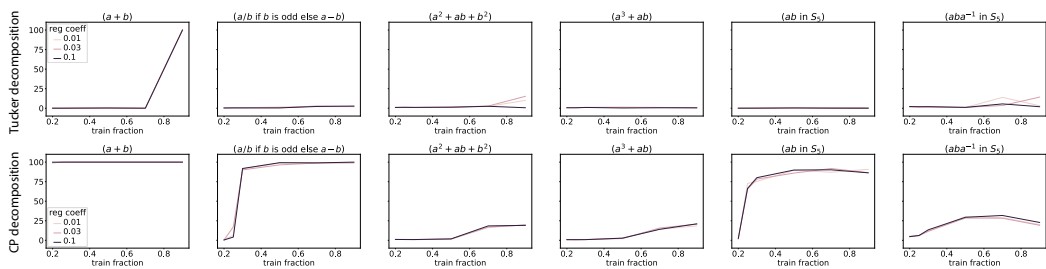

Figure 11: **Alternative Tensor Factorization Methods**: Test accuracy of (Top) Tucker and (Bottom) CP decomposition methods, trained across a range of $L_2$ regularization strengths.

---

[5]CP decomposition can also be viewed as a special case of Tucker decomposition with a fixed core tensor

$$M_{ijk} = 1 \quad \text{if } i = j = k, \quad 0 \quad \text{otherwise.}$$

# E    BAND-DIAGONAL HYPERCUBE

As mentioned above, HyperCube with diagonal embeddings lacks the capacity to effectively capture general group structures. However, the regular representation of a group generally decomposes into a direct sum of smaller irreducible representations, resulting in a sparse, block-diagonal matrix structure. Such block-diagonal structure can be effectively captured within the parameter space of *band-diagonal* matrices.

Therefore, to enhance the scalability of HyperCube, we explore the band-diagonal variant where the factor matrices are constrained to have a fixed bandwidth around the diagonal. This reduces the model's parameter count from $\mathcal{O}(n^3)$ to $\mathcal{O}(n^2)$, offering significant computational advantages.

Figure 12 compares the performance of the full HyperCube and the band-diagonal HyperCube with a bandwidth of 8 on a subset of tasks from Appendix B ($n = 97$ or $120$). Remarkably, the band-diagonal version exhibits comparable performance to the full HyperCube model, demonstrating its effectiveness in capturing group structures even with a significantly reduced number of parameters. This result highlights the potential of band-diagonal HyperCube for scaling to larger problems.

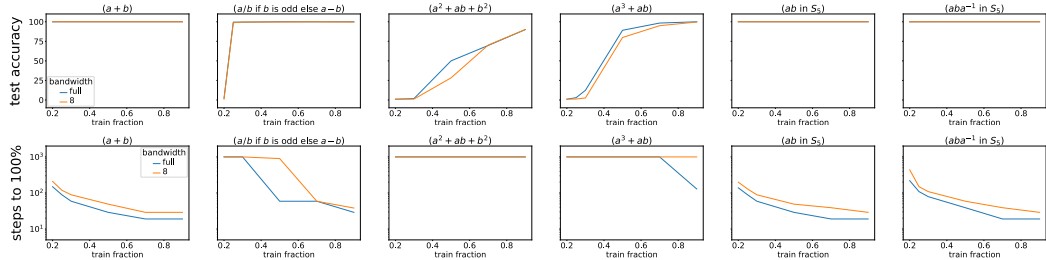

Figure 12: Full HyperCube vs Band-diagonal HyperCube model. (Top) final test accuracy, and (Bottom) steps to 100% test accuracy. lr = 0.5, reg coeff $\epsilon = 0.05$, init scale = 1.0.

# F    RUN-TIME COMPLEXITY

We empirically evaluate the run-time complexity of HyperCube. As expected, CPU execution time scales as $O(n^3)$. However, due to the efficient parallelization of einsum operations in PyTorch (See Appendix A), GPU execution time remains nearly constant with increasing $n$ (up to $n = 200$, the maximum size that fits in the 16GB memory of a Tesla V100 GPU). This demonstrates the practical efficiency of HyperCube when leveraging GPU acceleration.

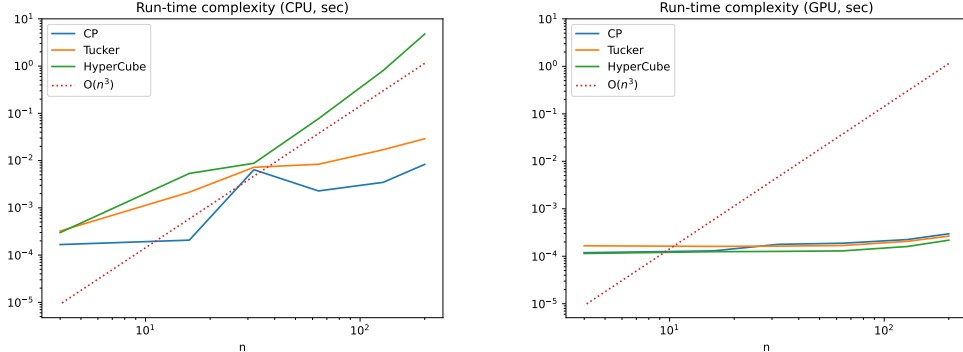

Figure 13: **Run-time complexity** for computing the HyperCube architecture (eq (4)) as functions of n. Other tensor decomposition methods (CP and Tucker) are also shown. (Left) Run-time on CPU. (Right) Run-time on GPU (Tesla V100 16GB). Results are averaged over 10 runs.

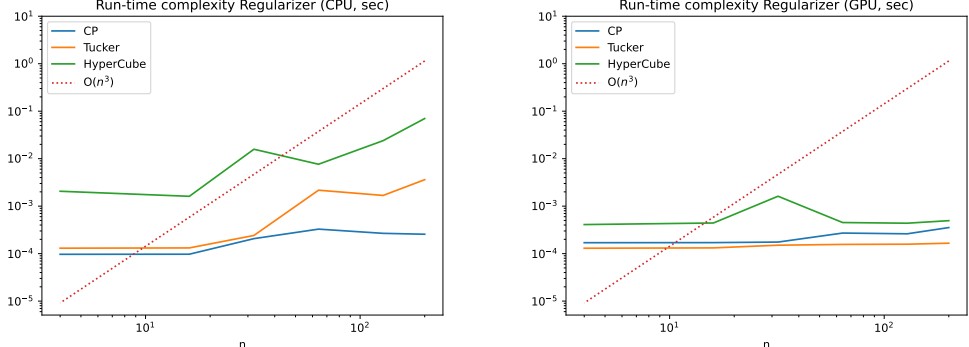

Figure 14:   Same as above but for computing the regularizers.

# G DEFERRED PROOFS

## G.1 PROOF OF LEMMA 5.1 ON BALANCED CONDITION OF HYPERCUBE

Here, we derive the balanced condition eq (7). The gradient of the regularized loss $\mathcal{L} = \mathcal{L}_o(T; D) + \epsilon \mathcal{H}(A, B, C)$ is

$$\nabla_{A_a}\mathcal{L} = \frac{1}{n}((\nabla_{T_{abc}}\mathcal{L}_o)\,C_c^\dagger B_b^\dagger + 2\epsilon(A_a(B_b B_b^\dagger) + (C_c^\dagger C_c)A_a)), \tag{14}$$

$$\nabla_{B_b}\mathcal{L} = \frac{1}{n}((\nabla_{T_{abc}}\mathcal{L}_o)\,A_a^\dagger C_c^\dagger + 2\epsilon(B_b(C_c C_c^\dagger) + (A_a^\dagger A_a)B_b)),$$

$$\nabla_{C_c}\mathcal{L} = \frac{1}{n}((\nabla_{T_{abc}}\mathcal{L}_o)\,B_b^\dagger A_a^\dagger + 2\epsilon(C_c(A_a A_a^\dagger) + (B_b^\dagger B_b)C_c)),$$

where $\nabla_{A_a}\mathcal{L} \equiv \partial\mathcal{L}/\partial A_a$, $\nabla_{B_b}\mathcal{L} \equiv \partial\mathcal{L}/\partial B_b$, $\nabla_{C_c}\mathcal{L} \equiv \partial\mathcal{L}/\partial C_c$, and $\nabla_{T_{abc}}\mathcal{L}_o \equiv \partial\mathcal{L}_o/\partial T_{abc}$.

Define the *imbalances* as the differences of loss gradients:

$$\xi_I \equiv \frac{n}{2\epsilon}(A_a^\dagger(\nabla_{A_a}\mathcal{L}) - (\nabla_{B_b}\mathcal{L})B_b^\dagger) = A_a^\dagger(C_c^\dagger C_c)A_a - B_b(C_c C_c^\dagger)B_b^\dagger$$

$$\xi_J \equiv \frac{n}{2\epsilon}(B_b^\dagger(\nabla_{B_b}\mathcal{L}) - (\nabla_{C_c}\mathcal{L})C_c^\dagger) = B_b^\dagger(A_a^\dagger A_a)B_b - C_c(A_a A_a^\dagger)C_c^\dagger$$

$$\xi_K \equiv \frac{n}{2\epsilon}(C_c^\dagger(\nabla_{C_c}\mathcal{L}) - (\nabla_{A_a}\mathcal{L})A_a^\dagger) = C_c^\dagger(B_b^\dagger B_b)C_c - A_a(B_b B_b^\dagger)A_a^\dagger$$

Setting the gradient to zero yields the balanced condition at stationary points, $\xi_I = \xi_J = \xi_K = 0$, which proves Lemma 5.1. Note that imbalance terms are defined to cancel out the $\nabla_{T_{abc}}\mathcal{L}_o$ terms. Therefore, the balanced condition is independent of the loss function $\mathcal{L}_o$.

## G.2 PROOF OF LEMMA 5.4

*Proof.* The constraint on Frobenius norm can be integrated with the regularizer into an augmented loss via the Lagrange multiplier $\lambda$

$$\mathcal{H} + \lambda(\mathcal{F} - constant), \tag{15}$$

where $\mathcal{F} \equiv \frac{1}{n} \text{Tr}\left[A_a^\dagger A_a + B_b^\dagger B_b + C_c^\dagger C_c\right]$ is the Frobenius norm .

The gradient of eq (15) with respect to $A_a$ is proportional to

$$\nabla_{A_a}(\mathcal{H} + \lambda\mathcal{F}) \propto A_a(B_b B_b^\dagger) + (C_c^\dagger C_c)A_a + \lambda A_a. \tag{16}$$

In the case of C-unitary factors $B$ and $C$, all terms in eq (16) become aligned to $A_a$, *i.e.*

$$\nabla_{A_a}(\mathcal{H} + \lambda\mathcal{F}) \propto (\alpha_B^2 + \alpha_C^2 + \lambda)A_a. \tag{17}$$

and thus an appropriate value for the Lagrange multiplier $\lambda$ can be found to vanish the gradient, which confirms stationarity. This result also applies to gradient with respect to $B_b$ and $C_c$ by the symmetry of parameterization. $\square$

## G.3 PERSISTENCE OF GROUP REPRESENTATION

The following lemma demonstrates a key property of our model's convergence behavior: once a group representation is learned, the solution remains within this representational form throughout optimization.

**Lemma G.1.** *Let $D$ represent a group operation table. Once gradient descent of the regularized loss eq (5) converges to a group representation (including scalar multiples), i.e.*

$$A_a = \alpha_{A_a}\varrho(a), \; B_b = \alpha_{B_b}\varrho(b), \; C_c = \alpha_{C_c}\varrho(c)^\dagger, \tag{18}$$

*the solution remains within this representation form.*

*Proof.* For the squared loss

$$\mathcal{L}_o(T; D) = \sum_{(a,b,c) \in \Omega_{\text{train}}} (T_{abc} - D_{abc})^2, \tag{19}$$

the gradient with respect to $A_a$ eq (14) becomes

$$\nabla_{A_a} \mathcal{L} = \frac{1}{n}(\Delta_{abc} M_{abc} C_c^\dagger B_b^\dagger + \epsilon(A_a(B_b B_b^\dagger) + (C_c^\dagger C_c)A_a)) \tag{20}$$

where $\Delta \equiv T - D$ is the constraint error, and $M$ is the mask indicating observed entries in the train set.

Substituting the group representation form eq (18) into eq (20), we get:

$$\frac{1}{n}\epsilon(A_a(B_b B_b^\dagger) + (C_c^\dagger C_c)A_a) = 2\epsilon\alpha_{A_a}\alpha^2 \varrho(a), \tag{21}$$

for the last two terms, where $\alpha^2 = \sum_b \alpha_{B_b}^2/n = \sum_c \alpha_{C_c}^2/n$.

Since the product tensor is

$$T_{abc} = \frac{1}{n} \text{Tr}[A_a B_b C_c] = \frac{1}{n}\alpha_{A_a}\alpha_{B_b}\alpha_{C_c} \text{Tr}[\varrho(a)\varrho(b)\varrho(c)^\dagger] = \alpha_{A_a}\alpha_{B_b}\alpha_{C_c} D_{abc},$$

and $D_{abc} = \delta_{a \circ b, c} = \delta_{a, c \circ b^{-1}}$ ($\delta$ is the Kronecker delta function), the first term in eq (20) becomes

$$\frac{1}{n}\sum_{b,c} \Delta_{abc} M_{abc} C_c^\dagger B_b^\dagger = \frac{1}{n}\sum_{b,c} \delta_{a \circ b, c} M_{abc}(\alpha_{A_a}\alpha_{B_b}\alpha_{C_c} - 1)\alpha_{B_b}\alpha_{C_c}\varrho(c \circ b^{-1})$$

$$= \frac{1}{n}\sum_b M_{ab(a \circ b)}(\alpha_{A_a}\alpha_{B_b}\alpha_{C_{a \circ b}} - 1)\alpha_{B_b}\alpha_{C_{a \circ b}}\varrho(a). \tag{22}$$

Note that both eq (22) and eq (21) are proportional to $\varrho(a)$. Consequently, we have $\nabla_{A_a}\mathcal{L} \propto \varrho(a)$. Similar results for other factors can also be derived: $\nabla_{B_b}\mathcal{L} \propto \varrho(b)$, and $\nabla_{C_c}\mathcal{L} \propto \varrho(c)^\dagger$. This implies that gradient descent preserves the form of the group representation (eq (18)), only updating the coefficients $\alpha_{A_a}, \alpha_{B_b}, \alpha_{C_c}$. $\square$

**Effect of $\epsilon$-Scheduler** Lemma G.1 holds true even when $\epsilon$ gets modified by $\epsilon$-scheduler, which reduces $\epsilon$ to 0. In this case, the coefficients converge to $\alpha_{A_a} = \alpha_{B_b} = \alpha_{C_c} = 1$, resulting in the exact group representation form eq (9).

# H GROUP CONVOLUTION AND FOURIER TRANSFORM

## H.1 FOURIER TRANSFORM ON GROUPS

The Fourier transform of a function $f : G \to \mathbb{R}$ at a representation $\varrho : G \to \mathrm{GL}(d_\varrho, \mathbb{R})$ of $G$ is

$$\hat{f}(\varrho) = \sum_{g \in G} f(g) \varrho(g). \tag{23}$$

For each representation $\varrho$ of $G$, $\hat{f}(\varrho)$ is a $d_\varrho \times d_\varrho$ matrix, where $d_\varrho$ is the degree of $\varrho$.

## H.2 DUAL GROUP

Let $\hat{G}$ be a complete set indexing the irreducible representations of $G$ up to isomorphism, called the *dual group*, thus for each $\xi$ we have an irreducible representation $\varrho_\xi : G \to U(V_\xi)$, and every irreducible representation is isomorphic to exactly one $\varrho_\xi$.

## H.3 INVERSE FOURIER TRANSFORM

The inverse Fourier transform at an element $g$ of $G$ is given by

$$f(g) = \frac{1}{|G|} \sum_{\xi \in \hat{G}} d_{\varrho_\xi} \, \mathrm{Tr} \left[ \varrho_\xi(g^{-1}) \hat{f}(\varrho_\xi) \right]. \tag{24}$$

where the summation goes over the complete set of irreps in $\hat{G}$.

## H.4 GROUP CONVOLUTION

The convolution of two functions over a finite group $f, g : G \to \mathbb{R}$ is defined as

$$(f * h)(c) \equiv \sum_{b \in G} f\left(c \circ b^{-1}\right) h(b) \tag{25}$$

## H.5 FOURIER TRANSFORM OF GROUP CONVOLUTION

Fourier transform of a convolution at any representation $\varrho$ of $G$ is given by the matrix multiplication

$$\widehat{f * h}(\varrho) = \hat{f}(\varrho)\hat{h}(\varrho). \tag{26}$$

In other words, in Fourier representation, the group convolution is simply implemented by the matrix multiplication.

*Proof.*

$$\widehat{f * h}(\varrho) \equiv \sum_c \varrho(c) \sum_b f(c \circ b^{-1}) h(b) \tag{27}$$

$$= \sum_c \varrho(c) \sum_{a,b} f(a) h(b) \delta_{(a, c \circ b^{-1})} \tag{28}$$

$$= \sum_{a,b} f(a) h(b) \sum_c \varrho(c) \delta_{(a \circ b, c)} \tag{29}$$

$$= \sum_{a,b} f(a) h(b) \varrho(a \circ b) \tag{30}$$

$$= \sum_a f(a) \varrho(a) \sum_b h(b) \varrho(b) \tag{31}$$

$$= \hat{f}(\varrho)\hat{h}(\varrho). \tag{32}$$

where $\delta$ is the Kronecker delta function, and the equivalence between $a = c \circ b^{-1}$ and $a \circ b = c$ is used between the second and the third equality. $\qquad \square$

# I   GROUP CONVOLUTION AND FOURIER TRANSFORM IN HYPERCUBE

HyperCube shares a close connection with group convolution and Fourier transform. On finite groups, the Fourier transform generalizes classical Fourier analysis to functions defined on the group: $f : G \rightarrow \mathbb{R}$. Instead of decomposing by frequency, it uses the group's irreducible representations $\{\varrho_\xi\}$, where $\xi$ indexes the irreps (See Appendix H.2). A function's Fourier component at $\xi$ is defined as:

$$\hat{f}_\xi \equiv \sum_{g \in G} f(g) \varrho_\xi(g). \tag{33}$$

**Fourier Transform in HyperCube**   The Fourier transform perspective offers a new way to understand how HyperCube with a group representation eq (9) processes general input vectors. Consider a vector $f$ representing a function, *i.e.*, $f_g = f(g)$. Contracting $f$ with a model factor $A$ (or $B$) yields:

$$\hat{f} \equiv f_g A_g = \sum_{g \in G} f(g) \varrho(g), \tag{34}$$

which calculates the Fourier transform of $f$ using the regular representation $\varrho$. As $\varrho$ contains all irreps of the group, $\hat{f}$ holds the complete set of Fourier components. Conversely, contracting $\hat{f}$ with $\varrho^\dagger$ (*i.e.* factor $C$) performs the *inverse Fourier transform*:

$$\frac{1}{n} \operatorname{Tr}[\hat{f} C_g] = \frac{1}{n} \sum_{g' \in G} f_{g'} \operatorname{Tr}[\varrho(g') \varrho(g)^\dagger] = f_g, \tag{35}$$

where eq (2) is used. This reveals that the factor tensors generalize the discrete Fourier transform (DFT) matrix, allowing the model to map signals between the group space and its Fourier (frequency) space representations.

Through the lens of Fourier transform, we can understand how the model eq (10) processes general input vectors ($f$ and $h$): it calculates their Fourier transforms ($\hat{f}, \hat{h}$), multiplies them in the Fourier domain ($\hat{f}\hat{h}$), and applies the inverse Fourier transform. Remarkably, this process is equivalent to performing group convolution ($f * h$). This is because the linearized group operation (Section 4.1) naturally entails group convolution (see Appendix I.1,**??**).

This connection reveals a profound discovery: HyperCube's ability to learn symbolic operations is fundamentally the same as learning the core structure of group convolutions. This means HyperCube can automatically discover the essential architecture needed for equivariant networks, without the need to hand-design them. This finding highlights the broad potential of HyperCube's inductive bias, extending its applicability beyond the realm of symbolic operations.

## I.1   REINTERPRETING HYPERCUBE'S COMPUTATION

HyperCube equipped with group representation eq (10) processes general input vectors $f$ and $h$ as

$$
\begin{aligned}
f_a h_b T_{abc} &= \frac{1}{n} \sum_a \sum_b f(a) h(b) \operatorname{Tr} \left[ \varrho(a) \varrho(b) \varrho(c)^\dagger \right] \\
&= \frac{1}{n} \operatorname{Tr} \left[ \left( \sum_a \varrho(a) f(a) \right) \left( \sum_b \varrho(b) h(b) \right) \varrho(c)^\dagger \right] \\
&= \frac{1}{n} \operatorname{Tr}[(\hat{f}\hat{h}) \varrho(c)^\dagger] = \frac{1}{n} \operatorname{Tr}[\widehat{f * h}\, \varrho(c)^\dagger] \\
&= (f * h)_c.
\end{aligned} \tag{36}
$$

Therefore, the model calculates the Fourier transform of the inputs ($\hat{f}$ and $\hat{h}$), multiplies them in the Fourier domain ($\hat{f}\hat{h}$), and applies the inverse Fourier transform, which is equivalent to the group convolution, as shown in Appendix H.5.

## J    SUPPLEMENTARY FIGURES FOR SECTION 6

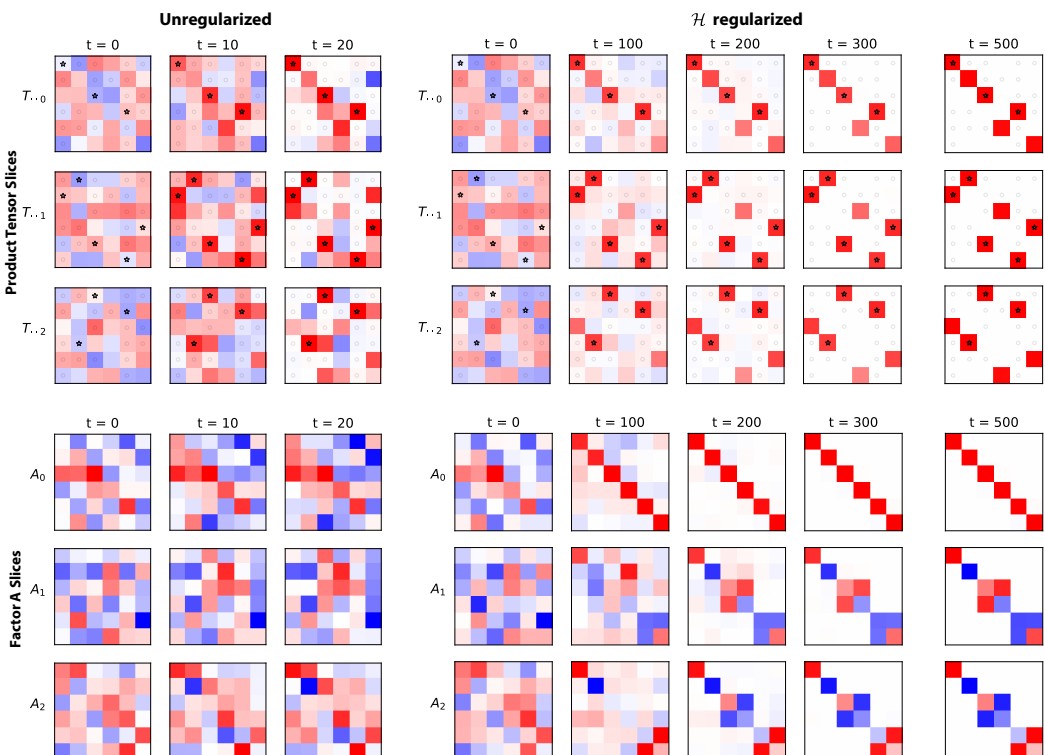

Figure 15:    Visualization of the end-to-end model tensor $T$ and the factor $A$ over the training iteration steps on the symmetric group $S_3$ task in Sec 6. Only the first three slices of the tensors are shown. (Top) End-to-end model tensor $T$: In the un-regularized case, the model tensor quickly converges to fit the observed data tensor entries in the training dataset (marked by stars and circles), but not in the test dataset. The $\mathcal{H}$-regularized model converges to a generalizing solution around $t = 200$. It accurately recovers $D$ when the regularization diminishes around $t = 400$ ($\epsilon \to 0$). (Bottom) Factor tensor $A$. The unregularized model shows minimal changes from random initial values, while $\mathcal{H}$-regularized model shows significant internal restructuring. Shown in the block-diagonalizing coordinate. See Fig 16 (Bottom). (color scheme: red=1, white=0, blue=-1.)

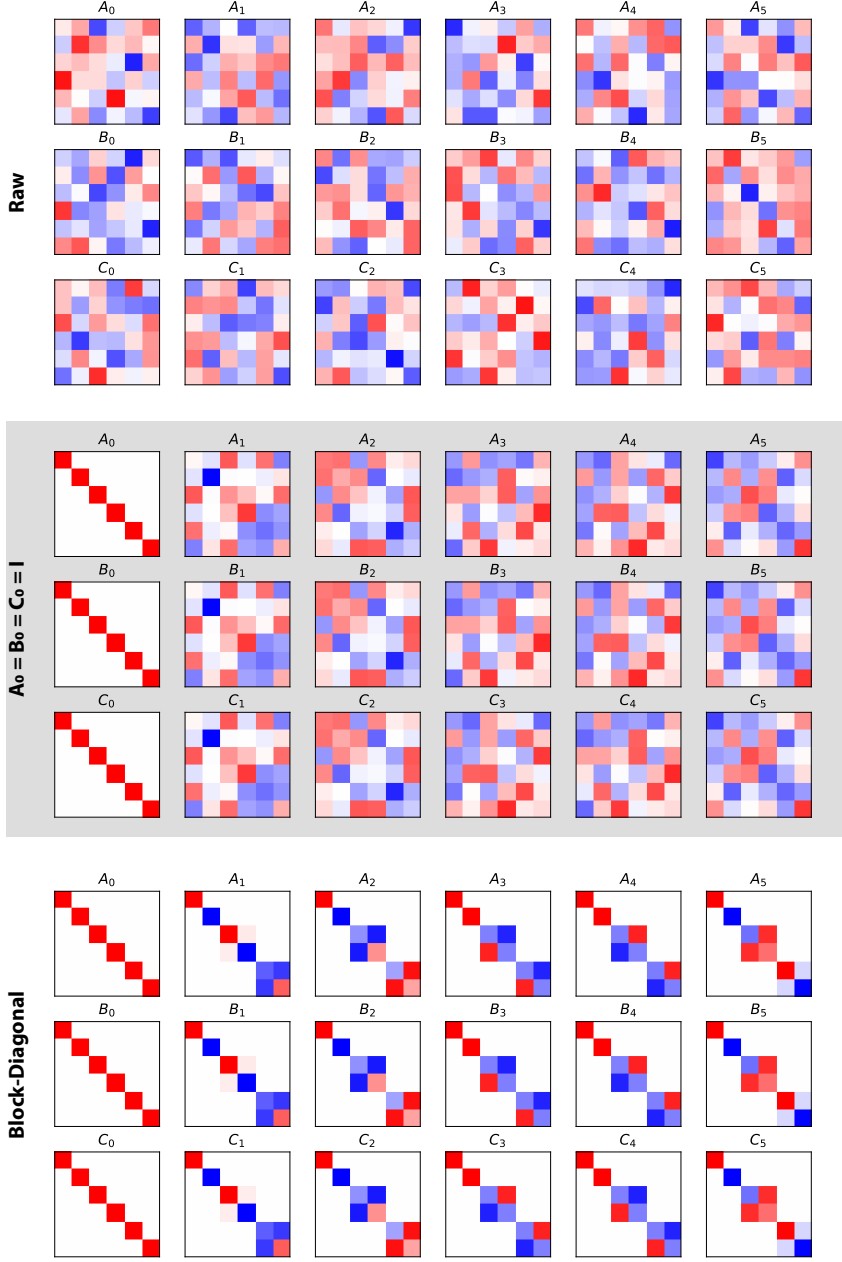

Figure 16: Learned factors of the $\mathcal{H}$ regularized model trained on the $S_3$ group. (Top) Raw factor weights shown in their native coordinate representation. (Middle) Unitary basis change as described in Sec 4.4 with $M_I = I$, $M_K = A_0$, $M_J = B_0^\dagger$, such that $\tilde{A}_0 = \tilde{B}_0 = \tilde{C}_0 = I$. Note that the factors share same weights (up to transpose in factor $\tilde{C}$). (Bottom) Factors represented in a block-diagonalizing basis coordinate, revealing the decomposition into direct sum of irreducible representations (irreps). (color scheme: red=1, white=0, blue=-1.)

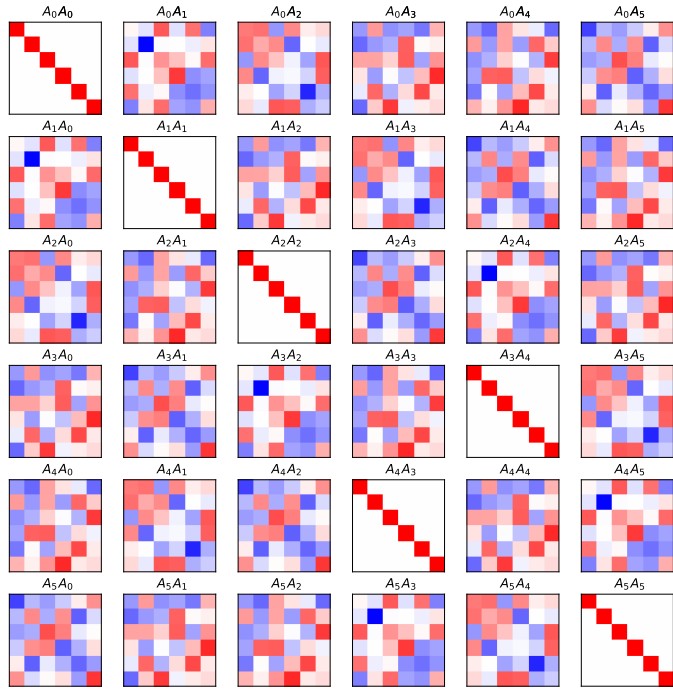

Figure 17: Multiplication table of matrix slices of factor $A$ from the mid panel of Fig 16. Note that this table share the same structure as the Cayley table of the symmetric group $S_3$ in Fig 2A. (color scheme: red=1, white=0, blue=-1.)

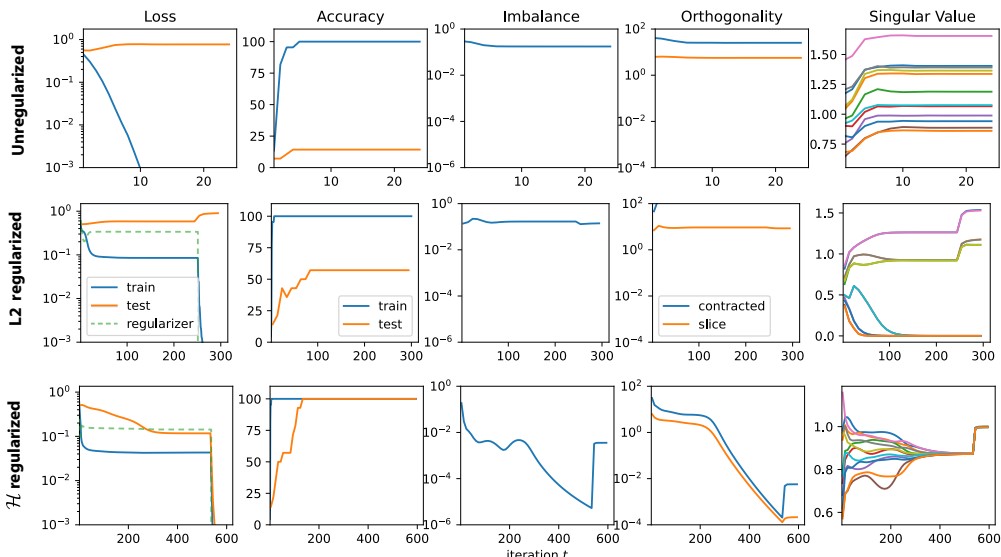

Figure 18: Optimization trajectories on the modular addition (cyclic group $C_6$) dataset, with 60% of the Cayley table used as train dataset (see Fig 19). (Top) Unregularized, (Middle) $L_2$-regularized, and (Bottom) $\mathcal{H}$-regularized training. The $L_2$-regularized model only achieves $\sim$60% test accuracy.

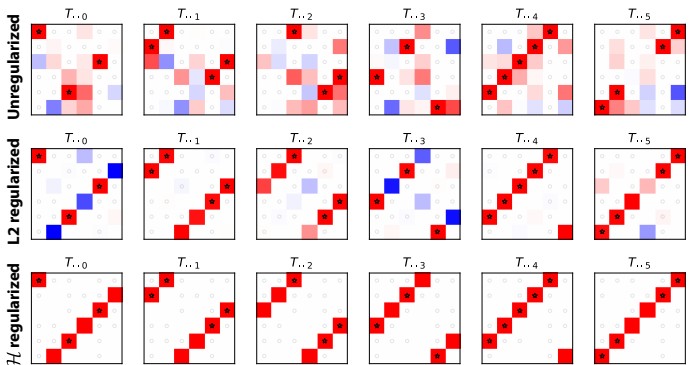

Figure 19: Visualization of end-to-end model tensor $T$ trained on the modular addition (cyclic group $C_6$) under different regularization strategies (see Fig 18). The observed training data are marked by asterisks (1s) and circles (0s). Only the $\mathcal{H}$-regularized model perfectly recovers the data tensor $D$. (color scheme: red=1, white=0, blue=-1.)

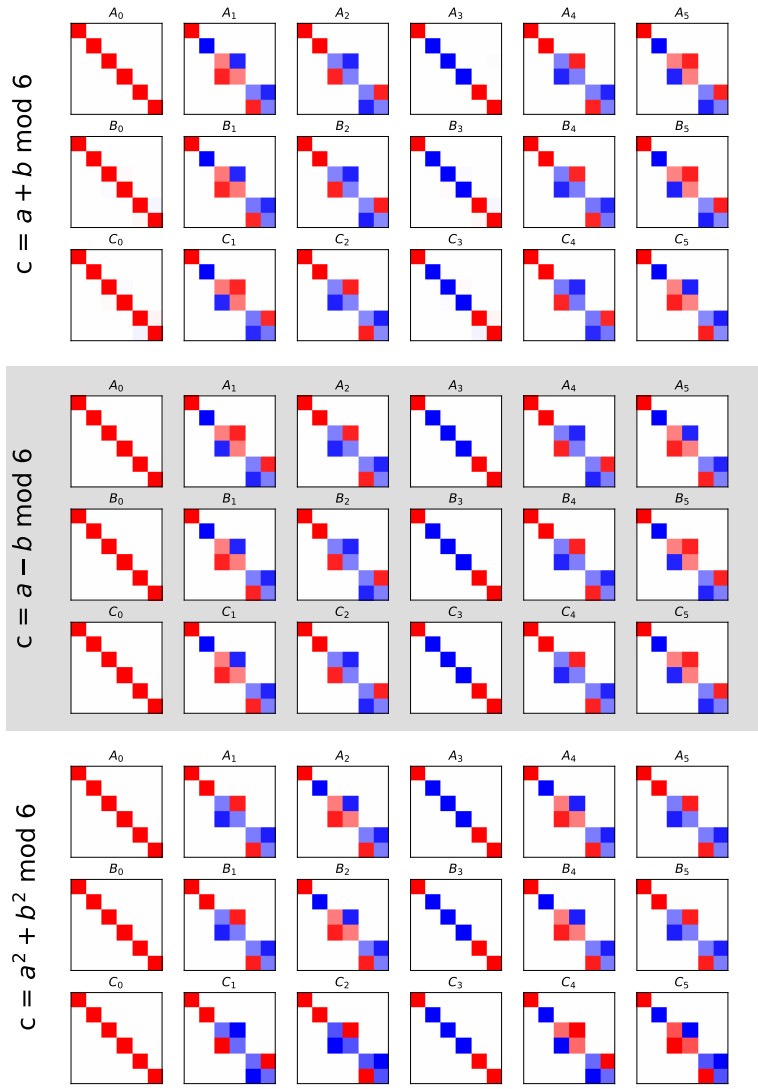

Figure 20: Visualization of factors trained on small Cayley tables from Figure 2. (Top) $c = a + b$ mod 6, satisfying $A_g = B_g = C_g^\dagger = \varrho(g)$. (Middle) $c = a - b$ mod 6, satisfying $A_g^\dagger = B_g = C_g = \varrho(g)$. (Bottom) $c = a^2 + b^2$ mod 6, which exhibits the same representation as modular addition for elements with unique inverses (e.g., $g = 0, 3$). For others, it learns *duplicate* representations reflecting the periodicity of squaring modulo 6: *e.g.*, $A_2 = A_4$ and $A_1 = A_5$, since $2^2 = 4^2$ and $1^2 = 5^2$. (color scheme: red=1, white=0, blue=-1.)

