# OpenReview forum: "Discovering Group Structures via Unitary Representation Learning"
_ICLR.cc/2025/Conference — ICLR 2025 Poster_

### Official Review · Reviewer_SQSt · 2024-10-27

**Soundness:** 3
**Presentation:** 3
**Contribution:** 3
**Rating:** 6
**Confidence:** 3

**Summary:**

The paper introduces a novel differentiable approach for discovering group structures within data by leveraging group representation theory. Traditionally, the non-differentiable nature of group axioms has posed challenges for their integration into deep learning frameworks. The proposed method addresses this limitation by employing a tensor-factorization model with matrix embeddings for each group element, along with a regularizer that encourages the learning of unitary matrix representations. This setup provides a strong inductive bias toward capturing group structures in data. The model is evaluated using Symbolic Operation Completion (SOC) tasks, where it successfully learns group operations from limited data and accurately discovers unitary representations. Furthermore, the model introduces an implicit complexity metric, facilitating the discovery of group structures in a variety of datasets, with potential applications in areas like symmetry discovery.

The method builds on well-established principles from group representation theory but extends them into a differentiable framework, making it applicable in modern deep learning contexts. By transforming SOC tasks into a tensor completion problem, the authors present a linearized framework that captures binary operations through bilinear maps, with a novel architecture—Hypercube—used to factorize the tensors. The accompanying regularizer plays a critical role in promoting unitary representations, ensuring that the model adheres to group properties.

**Strengths:**

- Differentiable Framework for Group Discovery: A notable strength of the paper is its introduction of a differentiable method for discovering group structures, addressing a long-standing challenge in integrating group theory with deep learning. By employing tensor-factorization and matrix embeddings, along with a regularizer promoting unitary matrices, the approach allows for the learning of group representations within a differentiable framework. This innovation enhances the applicability of group theory in machine learning, providing a more seamless way to incorporate group structures into data-driven models, particularly in contexts requiring automatic discovery of algebraic properties.
- Solid Theoretical Foundation with Practical Relevance: Another strength lies in the paper's solid grounding in group representation theory, combined with its practical application to SOC tasks. The method’s reliance on well-established theoretical principles, such as the use of a complexity metric and regularization to guide learning, is complemented by its practical effectiveness in recovering group operations and unitary representations from limited data. This balance between theory and practice makes the approach both rigorous and relevant, showcasing its potential for broader applications in tasks involving the discovery of group structures within data.

**Weaknesses:**

- Narrow Focus on SOC Tasks: One of the paper's weaknesses is its narrow focus on SOC tasks for evaluation. While SOC tasks provide a controlled environment to test group structure discovery, they may not fully capture the model’s performance or versatility in real-world applications, where data and tasks are often more varied and complex. By limiting the scope to SOC tasks, the paper does not explore how well the proposed method generalizes to other tasks like graph classification, natural language processing, or scientific data analysis, which could broaden the understanding of its practical impact.
- Limited Baseline Comparisons: The comparison of Hypercube to other models, especially the Transformer from Power et al. (2022), is somewhat limited. While the paper shows that Hypercube outperforms the Transformer in terms of learning speed, the focus on just one baseline model limits the robustness of the evaluation. There is no detailed comparison against other relevant methods that also target group discovery or structure learning. Including a wider range of baseline comparisons would provide a more comprehensive assessment of Hypercube's relative strengths and weaknesses.

**Questions:**

The paper does not provide much detail on how sensitive the performance of HyperCube is to various hyperparameters, such as the regularization strength, learning rate, or factor initialization. Understanding the impact of these hyperparameters, especially across different tasks (group vs. non-group operations), would be useful for replicating and extending the work to new datasets and applications. Can the authors provide more clarity on the tuning process and how these parameters affect the model's performance?

---

> ### Author Response · Authors · 2024-11-21
> **Response to Reviewer SQSt**
>
> We thank the reviewer for their thoughtful feedback and appreciate the recognition of our work's strengths, including the novel differentiable framework for group discovery and its solid theoretical foundation. We address the concerns raised below:
>
> **Minor change**: We replaced SOC (Symbolic Operation Completion) with BOC (Binary Operation Completion) for clarity.
>
> ### Narrow Focus on BOC Tasks:
> While we agree on the importance of broader evaluation, the focus on binary operations is a deliberate choice with significant advantages. Since a group is defined its binary operation, BOC allows us to isolate the core challenge of learning group structures for rigorous analysis, without the confounding influence of other factors. This helps establish a theoretical foundation for structure learning in discrete symbolic operations, akin to how matrix completion serves as foundational analysis in the continuous domain.
>
> We are actively working on extending this work to more complex tasks and applications, building upon the foundation established here. For example, we are extending this approach to more complex scenarios, such as discovering the convolutional structure of equivariant neural networks, which involves simultaneously learning the group structure and kernel weights. By first focusing on structure learning in isolation, we provide a clear and rigorous analysis essential for understanding more complex settings.
>
> Furthermore, we believe our work has implications for scientific data and application domains where a clear mathematical notion of underlying symmetry structure is lacking or non-exact. For example, natural language exhibits semantic symmetries (e.g., paraphrasing), but the complete structure of these symmetries remains poorly understood. Our approach may offer new tools to analyze and leverage such structures.
>
> ### Limited Baseline Comparisons:
> We acknowledge the limitation of having a single primary baseline; however, directly comparable prior work is scarce. To the best of our knowledge, no other work addresses group structure learning from purely symbolic relationships. As noted in the Related Works section, existing symmetry discovery methods often assume additional information about the underlying group structure, making them unsuitable for direct comparison. This scarcity underscores the novelty of our approach and highlights the need for further research in this area.
>
> ### Hyperparameter Sensitivity:
> We now include hyperparameter sensitivity analysis in the Appendix D. As described in the General Response, HyperCube demonstrates robustness across a wide range of hyperparameter values.  Learning rate and regularization strength primarily affect convergence speed without impacting final accuracy, though extreme values can lead to instability. Initialization scale has no significant effect. Importantly, the same hyperparameter settings generalize well across all tasks (both group and non-group operations), suggesting that HyperCube can be readily applied to new problems without extensive tuning.

---

### Official Review · Reviewer_13RW · 2024-10-29

**Soundness:** 2
**Presentation:** 3
**Contribution:** 1
**Rating:** 6
**Confidence:** 4

**Summary:**

This paper studies a method to predict the complete group elements based on partial observations of the action of a finite group using tensor decomposition. The HyperCube decomposition and the regularization term presented in equation (6) are newly proposed. This regularization term theoretically promotes the "Imbalance" and "Unitarity" of the decomposed factor parameters. Experiments are conducted on data involving simple operations like integer addition and subtraction, comparing the performance with a Transformer.

**Strengths:**

- The idea of connecting group theory to tensor decomposition to solve this problem is unique and interesting.
- The paper is written in a clear and accessible way, starting from fundamental concepts.

**Weaknesses:**

- The necessity of the inductive bias introduced in this work is not adequately explained, particularly why this specific bias was chosen over others.
- The proposed method does not scale well. If the number of group elements is n, the computational complexity is O(n^3).
- The experimental results mentioned in the text (Figures 10 and 11) are pushed to the appendix, potentially violating the page limit.

**Questions:**

- The paper solves the problem of predicting group actions, but how does this lead to symmetry discovery? Could the authors provide concrete ideas?
- How could this method be adapted to continuous groups, such as Lie groups?
- Could the authors explain in what criterion the inductive bias introduced by (5) was chosen? For example, the tensor T[i, j, :] is strongly constrained to become a one-hot vector for all i, j. Could another approach, like a classification model where $y_{ijk} = \langle A_i, C_k, B_j \rangle$ (with parameters $A_i, B_j \in \mathbb{R}^k $ and $ C_k \in \mathbb{R}^{k \times k}$ where k is an embedding dimension) serve as a better model? What advantages does the proposed inductive bias offer compared to this alternative?
- The authors claim that the sample complexity is better than that of a Transformer, but what about the computational time (wallclock) in comparison?
- Could the authors provide a more intuitive explanation of the "Imbalanceness" in Lemma 5.1?
- How sensitive is the prediction accuracy to $ \epsilon $?

---

> ### Author Response · Authors · 2024-11-22
> **Response to Reviewer 13RW**
>
> Thank you for your constructive feedback. We address your specific concerns and questions below:
>
> ### Necessity of HyperCube's Inductive Bias:
> As detailed in the General Response, HyperCube's design helps directly encoding the *associativity axiom* of groups in its architecture and leverages the *Unitarity Theorem* through the regularizer to significantly *reduce the space of possible solutions* without loss of generality. This focused search space promotes faster convergence and enhances generalizability. This is now clearly explained in the improved Section 4.
> To further demonstrate the advantage of this inductive bias, Appendix F provides a comparison to other tensor factorization architectures and discusses their limitations.
>
> ### Scalability:
> We have addressed the scalability concern in the conclusion (scalability paragraph) and the new Appendix E, as described in the General Response.
>
> ### Placement of Figures:
> The reviewer noted that Figures 10 and 11 are in the appendix, potentially violating the page limit. These visualizations are intended as supplementary aids to help readers understand the structure of learned representations, rather than serving as primary results like Figures 5, 6, and 7. It is common practice to include such figures in supplementary materials, especially when they are large and space-consuming.  While we acknowledge the page limit constraint, including these figures in the main text would disrupt the flow and readability of the paper.
>
> ### Symmetry Discovery:
> Our method offers a novel approach to symmetry discovery within the framework of geometric deep learning. In geometric deep learning, equivariant layers are typically implemented using bilinear layers of the form  $o_c = T_{abc} x_a w_b$ where $x$ and $w$ are the input vectors, $o$ is the output vector, and $T$ is a predefined tensor representing a specific group convolution corresponding to a known symmetry. Here, $w$ serves as the learnable weight vector (i.e., the convolution kernel).
>
> In contrast, our approach allows both $T$ and $w$ to be learned simultaneously, while constraining $T$ to satisfy group axioms in a differentiable manner. This enables the discovery of potentially novel group structures and their corresponding symmetries directly from the data, without relying on predefined symmetries. Appendix J provides a general setup for how $T$ mediates the group convolution over functions (vectors) defined on group elements. We are actively pursuing this research direction in our subsequent work.
>
> ### Adaptation to Continuous Groups:
> While the current method does not directly apply to continuous groups over an infinite set, the structure of a Lie group *G* is fully captured by its *Lie algebra*, a finite-dimensional vector space spanned by *infinitesimal generators*.
>
> Our method can be adapted to capture the axioms of Lie algebra by leveraging its representation theory. For example, Lie algebras satisfy the *Jacobi identity* instead of associativity. This property is reflected in the matrix representation of Lie algebra through the **commutator**: $[X,Y]=XY−YX$. Therefore, HyperCube can be adapted to incorporate the commutator in place of matrix multiplication within its architecture. We are currently exploring this direction.
>
> ### Rationale Behind HyperCube Factorization:
> As described in the General Response, HyperCube's factorization is grounded in representation theory, utilizing matrix embeddings and matrix multiplication to naturally encode the associativity axiom of groups within the architecture.
>
> The suggested factorization scheme appears to be a variant of Tucker decomposition where one of the matrix factors is replaced by an identity matrix:
> $T_{abc} =  \sum_{i,j,k} M_{ijk} A_{ai} B_{bj} \delta_{ck}$.
> This scheme shares the same limitations as Tucker decomposition, which are discussed in the General Response and Appendix F. Primarily, it struggles to effectively capture the algebraic relationships within the tensor due to the challenges associated with regularizing the core tensor, M. This limitation hinders its ability to learn and represent group structures accurately.

---

> > ### Author Response · Authors · 2024-11-22
> > **continued**
> >
> > * **One-hot Vector Constraint**:
> > The reviewer commented that the vector slices of the model tensor $T$ are strongly constrained to become one-hot vectors. While HyperCube's ability to learn and represent group structure allows it to naturally produce the correct one-hot outputs, it does not impose such a constraint.
> > * For example, the data tensor can be transformed by arbitrary unitary matrices ($U$s) as
> > $\tilde D_{a'b'c'} = D_{abc} U^A_{aa'} U^B_{bb'} U^C_{cc'}$,
> > and HyperCube can recover $\tilde D$ just as easily as $D$.
> > This demonstrates that the model's performance is not reliant on a specific one-hot constraint but rather on its ability to learn the underlying group structure.
> >
> > ### Run-Time Complexity Comparison:
> > As described in the General Response, despite the $O(n^3)$ scaling of compute and memory requirements, HyperCube runs efficiently on GPUs, exhibiting near-constant runtime (Appendix F). Training HyperCube on a single BOC task typically takes seconds to minutes on a V100 GPU, while training Transformers can take hours.
> > Figure 7 demonstrated that HyperCube boasts significantly faster (100x-1000x) convergence speed than Transformers in terms of training steps, in addition to better sample efficiency. This extremely slow convergence of Transformers was a key observation in Power et al. (2022).
> >
> >
> > ### Intuitive Explanation of "Balancedness":
> > Intuitively, balancedness describes a phenomenon in multi-part systems where, at optimality, the parts share a certain quantity. A classic example is a gas at thermal equilibrium, where uniform pressure and temperature are observed across the system.
> >
> > In deep learning, a well known example of this concept manifests in deep linear networks (and networks with homogeneous nonlinearities like ReLU) under L2 regularization. At stationary points, weight matrices become balanced, sharing singular values and vectors between adjacent layers. This balanced condition ultimately leads to a bias toward low-rank solutions.
> >
> > The balanced condition in Equation (7) is more complex due to the intricate regularizer and tensor architecture. However, it also ultimately leads to a shared quantity in the model: factors sharing the same unitary representations
> > (Under L2 regularization, this condition simplifies to
> > $A_a^\intercal A_a = B_{b} B_b^\intercal$,
> > $B_b^\intercal B_b = C_{c} C_c^\intercal$,
> > and $C_c^\intercal C_c =  A_{a} A_a^\intercal$,
> > similar to the balanced condition in deep linear networks.)
> >
> > We utilize this condition to derive specific properties of the solution, such as unitarity (Section 5), and as an indicator of convergence towards optimality (Figure 5), which illustrates the close relationship between unitarity and balancedness in HyperCube.
> >
> > ### Sensitivity to regularization strength:
> > As described in the General Response, HyperCube shows robust performance to variations in regularization strength as well as in other hyperparameters (Appendix D).
> >
> > We believe these revisions and clarifications address your concerns and strengthen the manuscript. We hope you find our responses satisfactory.

---

> > > ### Comment · Reviewer_13RW · 2024-11-27
> > >
> > > Thank you for your response. Some of my concerns (sensitivity, balancedness, continuous group, inductive bias, runtime comparison) are resolved. I will raise my score.
> > >
> > > Follow-up question:
> > > * I appreciate the additional experiment to measure the wallclock time (Appendix E). For large-scale data, however, we may handle it with larger n, say n=1000 or 10000. Can you observe the same trend --- the GPU wallclock time remains constant --- on such larger ns?

---

> > > > ### Author Response · Authors · 2024-11-27
> > > >
> > > > Thank you for your positive feedback and for raising your score. We are pleased that our revisions addressed some of your concerns.
> > > >
> > > > Regarding your follow-up question about scalability to larger values of n:
> > > >
> > > > While we couldn't run larger simulations due to GPU memory limitations, we believe the observed trend of near-constant GPU wallclock time will continue for larger n. This is because the computational complexity of HyperCube is dominated by matrix multiplications, which are highly optimized for GPUs.
> > > >
> > > > However, we acknowledge the potential memory issues for very large n. The band-diagonal HyperCube method helps mitigate this, but currently, it doesn't fully utilize the efficient einsum operation. We believe this can be further optimized to improve memory efficiency and enable scaling to even larger datasets.
> > > >
> > > > We would be happy to address any further questions or remaining concerns you may have, including those related to HyperCube's inductive bias or figure placement.

---

### Official Review · Reviewer_wdaY · 2024-11-04

**Soundness:** 3
**Presentation:** 2
**Contribution:** 3
**Rating:** 5
**Confidence:** 3

**Summary:**

The paper presents a novel differentiable approach for discovering group structures in data using group representation theory and tensor-factorization models. It effectively learns group operations and unitary representations from limited data while defining a complexity metric that enhances group structure discovery, impacting various scientific domains and automatic symmetry discovery applications.

**Strengths:**

1. This paper proposes a differentiable approach to the automatic discovery of finite group structures through the learning of their underlying representations.

2. This paper introduces a novel regularization technique that encourages the matrices to maintain unitarity, thereby improving the preservation of group structures.

3. This paper offers a theoretical analysis of the inductive bias associated with HyperCube.

**Weaknesses:**

1. The clarity of the paper's writing is lacking, which makes it somewhat challenging to read. For example, the rationale for introducing HyperCube and its advantages are not adequately explained. Additionally, the underlying intuition behind HyperCube regularization is insufficiently articulated.

2. The paper does not include a comprehensive set of experiments. L2 regularization is relatively weak compared to other regularization techniques. To strengthen the paper's contributions, it is essential to compare HyperCube regularization with more robust regularization methods.

3. Given that HyperCube regularization involves matrix multiplication, it is important to present both the computation and runtime in the experimental results. This information would provide valuable insights into the efficiency of the proposed method.

**Questions:**

1.  In Line 185, can a and b mapped to vector embeddings or diagonal matrix embeddings?

2. Can the tensor T be parameterized using other tensor decomposition models?

---

> ### Author Response · Authors · 2024-11-22
> **Response to Reviewer wdaY**
>
> Thank you for your thoughtful feedback and for recognizing the novelty and potential impact of our work. We address your specific concerns below:
>
> ### Clarity on HyperCube design
>
> We have improved Section 4 to provide a clearer explanation of the deep connection between HyperCube's design and representation theory. As detailed in the General Response, we now emphasize how HyperCube encodes associativity in its architecture and leverages the unitarity theorem via regularization to significantly reduce the search space of possible solutions. This focused search space promotes faster convergence and enhances generalizability. Additionally, HyperCube regularizer is now expressed in a more interpretable form, highlighting its *duality* to the standard L2 regularizer.
>
> ### Alternative regularizers
> While we appreciate the suggestion to compare HyperCube regularization with more robust methods, our focus in Section 5 is to highlight the *qualitative* differences between these regularization schemes in the context of learning group structures. It shows that L2 regularization, with its bias towards low-rank solutions (which is dual to HyperCube regularizer's bias), is fundamentally unsuited for capturing the full-rank nature of group representations. This qualitative comparison serves to support the theoretical conclusions of our analysis. Furthermore, Section 7 provides a *quantitative* comparison to a stronger Transformer baseline with extensive hyperparameter tuning and various regularization techniques. Given the strong theoretical foundation of our work, grounded in the representation theory, we believe our experiments, along with the comparison to other factorization techniques in Appendix F, provide sufficiently conclusive results.
>
> ### Computational and run-time complexity
> As described in the General Response, this information is now included in the Conclusion (scalability paragraph) and Appendix E. Despite the $O(n^3)$ scaling of memory and computational requirements, the runtime complexity of HyperCube remains nearly constant due to efficient parallelization on GPUs. Furthermore, we introduce a band-diagonal variant of HyperCube that reduces the parameter count and complexity to $O(n^2)$ without compromising performance. This variant offers a promising approach for scaling to larger problems.
>
>
> ### Alternative tensor decompositions
> This is now addressed in the Appendix F, where we discuss *Tucker and CP decompositions* and demonstrate their limitations in capturing group structures.
>
> ### Using vector or diagonal embedding
> As explained in the General Response, using diagonal matrix embeddings in HyperCube is equivalent to a CP decomposition with vector embeddings.  However, this restricted setting significantly limits the model's expressive power.  Specifically, it can only represent commutative groups, and even then, requires the complex field $K=\mathbb{C}$ to do so.  To capture the full richness of group structures, including non-commutative groups, the full matrix embeddings employed in HyperCube are essential.
>
> We believe these revisions and clarifications address your concerns and strengthen the manuscript. We hope you find our responses satisfactory.

---

> > ### Comment · Reviewer_wdaY · 2024-11-27
> >
> > Thank you for your response. However, I still have a few concerns that remain unaddressed.
> >
> > 1. You mentioned that "Section 7 provides a quantitative comparison to a stronger Transformer baseline with extensive hyperparameter tuning and various regularization techniques." However, I couldn't locate the comparison of different regularizations in the text. Could you please specify which line includes this information?
> >
> > 2. It would enhance the analysis to include the run-time of other architectures and regularization methods in Figure 11 for a more comprehensive comparison.

---

> > > ### Author Response · Authors · 2024-11-27
> > >
> > > Thank you for your continued engagement and feedback. We appreciate you pointing out these areas for improvement.
> > >
> > > Regarding the description of hyperparameter tuning: We apologize for the lack of clarity in our previous response.  We have now revised the manuscript to explicitly address this at the beginning of Section 7, between lines 464-468. This addition provides a more precise reference to the detailed hyperparameter tuning methods for the Transformer baseline, as described in Power et al. (2022).
> > >
> > > Regarding the run-time comparison:  We acknowledge the value of including run-time information for other architectures and regularization methods.  To address this, we have updated Figure 11 and added Figure 12 to provide a more comprehensive comparison of run-time complexity. Figure 11 now demonstrates that while HyperCube factorization is the slowest on CPU, the run-times are comparable on GPU, with near-constant performance across different values of n. Similarly, Figure 12 shows near-constant run-time for computing the regularizers on GPU, though HyperCube is 2-3 times slower than CP and Tucker.
> > >
> > > We hope these revisions and additions adequately address your concerns. Please let us know if you have any further questions.

---

> > > > ### Comment · Reviewer_wdaY · 2024-11-28
> > > >
> > > > Thank you for your response.
> > > >
> > > > I apologize for any confusion caused by my earlier communication. In Section 6.1, you compare H-regularization to L2 regularization; however, it is important to note that L2 regularization may be considered relatively weak. Consequently, I recommend conducting a comparative analysis of H-regularization against stronger regularization methods. To facilitate this, you can substitute $H(A,B,C)$ in Equation (5) with alternative regularization methods. This can effectively highlight the advantages of H-regularization.

---

> > > > > ### Author Response · Authors · 2024-11-28
> > > > >
> > > > > We appreciate the reviewer's continued engagement and feedback.  The suggestion to compare HyperCube regularization to a *stronger* alternative appears to be based on the following premises, which we request to be clarified:
> > > > >
> > > > >   1. **Our current manuscript does not fully demonstrate the advantages of HyperCube regularization**. We believe the manuscript already provides a thorough analysis of HyperCube regularization's properties, specifically its unique ability to induce a bias towards learning group structures. This is supported by both theoretical analysis and empirical results, including the comparison to the state-of-the-art Transformer model in Section 7.
> > > > >   2. **The perceived gap in understanding stems from the comparison to *weak* L2 regularization.** We argue that the comparison to L2 serves a specific purpose: to highlight the fundamental *qualitative* difference in their inductive bias. L2 promotes *low-rank* solutions, generally desirable in many machine learning contexts but unsuitable for representing group structures. HyperCube regularization, on the other hand, promotes *unitarity*, essential for capturing the algebraic properties of groups. This comparison provides a qualitative understanding of these distinct biases, not a *stronger* vs. *weaker* judgment.
> > > > >   3. **There exist widely recognized *stronger* regularization methods better suited for discovering group structures in tensor factorization architectures.** To the best of our knowledge, we are not aware of any such regularization methods. We would appreciate it if the reviewer could clarify in what sense L2 regularization is *weak* and provide concrete examples or references to such stronger methods.
> > > > >   4. **A comparison to such a *strong* regularization would fill the perceived gap in understanding.** It is unclear how this comparison would contribute meaningfully to further understanding the properties and advantages of HyperCube regularization, given its already demonstrated effectiveness and unique inductive bias.
> > > > >
> > > > > We are committed to a thorough analysis and remain open to further comparisons if they are well-motivated and contribute meaningfully to understanding HyperCube regularization. However, we believe the existing theoretical analysis and experimental results, including the comparison to the state-of-the-art Transformer model, already provide compelling evidence for the effectiveness of HyperCube regularization in learning group structures.
> > > > >
> > > > > Therefore, we respectfully request that the reviewer reconsider the necessity of this comparison. If the reviewer believes the current analysis is insufficient, we would greatly appreciate specific clarification and guidance on the points mentioned above. With such guidance, we can ensure that any further analysis is relevant and strengthens the paper. We are happy to make further revisions to address any remaining concerns.

---

> > > > > > ### Comment · Reviewer_wdaY · 2024-11-29
> > > > > >
> > > > > > Thank you for your responses.
> > > > > >
> > > > > > 1. Numerous regularization methods have been developed for tensor decomposition, including Lp norm, Lp,q norm, trace norm and so on. A selection of these methods is outlined below.
> > > > > >
> > > > > > Zheng Y, Xu A B. Tensor completion via tensor QR decomposition and L2, 1-norm minimization[J]. Signal Processing, 2021, 189: 108240.
> > > > > >
> > > > > > Fan J, Ding L, Yang C, et al. Euclidean-Norm-Induced Schatten-p Quasi-Norm Regularization for Low-Rank Tensor Completion and Tensor Robust Principal Component Analysis[J]. arXiv preprint arXiv:2012.03436, 2020.
> > > > > >
> > > > > > Liu Y, Shang F, Jiao L, et al. Trace norm regularized CANDECOMP/PARAFAC decomposition with missing data[J]. IEEE transactions on cybernetics, 2014, 45(11): 2437-2448.
> > > > > >
> > > > > > Lacroix T, Usunier N, Obozinski G. Canonical tensor decomposition for knowledge base completion[C]//International Conference on Machine Learning. PMLR, 2018: 2863-2872.
> > > > > >
> > > > > > 2. In comparing HyperCube decomposition to CP decomposition and Tucker decomposition, it should be noted that HyperCube decomposition incorporates the regularization method $H(A,B,C)$, whereas CP decomposition does not employ such a method. Consequently, it is advisable to evaluate HyperCube decomposition alongside other decomposition techniques, both with and without regularization methods. Furthermore, many differences exist between the Transformer baseline and HyperCube, complicating the interpretation of experimental results regarding the effectiveness of specific components within HyperCube.

---

> ### Author Response · Authors · 2024-12-01
>
> We thank the reviewer for their detailed feedback and suggestions. Below, we address the points raised:
>
> **Applicability of Alternative Regularization Methods**:
> While the suggested methods (e.g., Lp norms, trace norm) are effective for promoting sparsity or low-rank structures in tensor decomposition, they do not align with the algebraic requirements of group theory or the architecture of HyperCube factorization. Our use of L2 regularization serves a specific purpose: to contrast its low-rank bias with the unitarity-promoting bias of HyperCube regularization, as discussed in Section 6.1. This comparison highlights their fundamental qualitative differences in inductive bias rather than a stronger vs. weaker evaluation.
>
> In our earlier response, we requested specific examples of "stronger regularization methods" relevant to group structure discovery. However, the suggested alternatives do not inherently support this goal. We respectfully reiterate our request for clarification or examples of regularizers better suited to uncovering group structures. Without this guidance, it remains unclear how further comparisons would enhance the understanding of HyperCube’s unique advantages.
>
> **Tucker and CP Decompositions**:
> In addition to experimental results, Appendix F describes that Tucker and CP decompositions fundamentally lack the inductive bias or capacity required for learning group structures, regardless of the applied regularizers. Consequently, such comparisons would not yield meaningful insights for this study.
>
> **Transformer Baseline Comparison**:
> We acknowledge that comparisons with the Transformer baseline may involve confounding factors due to different architectures. However, the comparison serves to benchmark HyperCube’s overall performance against the state-of-the-art model that is highly-turned for BOC tasks, as established by Power et al. (2022). Furthermore, detailed analyses in Section 6 isolate HyperCube’s contributions, showing how its unitarity bias uniquely drives generalization and representation learning.
>
> **Request for Additional Comparisons**:
> The request to evaluate HyperCube decomposition alongside other decomposition techniques, both with and without regularization, is unnecessary and would require an unreasonable amount of additional work. Our manuscript already provides both theoretical and empirical evidence to understand the advantages and properties of HyperCube regularization.
>
> Importantly, the reviewer has not provided a clear justification for how these additional comparisons would meaningfully contribute to understanding HyperCube’s effectiveness or unique inductive bias. In the absence of such clarification, we respectfully request that the reviewer reconsider the necessity of these comparisons.
>
> We appreciate the reviewer’s engagement and thoughtful critique. If our response has adequately resolved the concerns raised, we kindly ask the reviewer to consider reflecting this in an updated review score.

---

### Author Response · Authors · 2024-11-21
**General responses**

We sincerely thank all reviewers for their thoughtful and constructive comments, which have significantly enhanced the quality of our work. Based on your invaluable feedback, we conducted additional experiments and expanded our discussions to address your suggestions.
We summarize main improvements below:

### Understanding HyperCube's design
We've improved Section 4 to provide a clearer explanation behind the model design, emphasizing connection to group axioms and representation theory, a more intuitive form of the regularizer,  and comparison to other factorization schemes.

HyperCube’s use of matrix embeddings and multiplication is grounded in the representation theory of finite groups, where the associative property of matrix multiplication naturally encodes the **associativity axiom** of groups. This provides a crucial advantage for representing groups effectively.

HyperCube regularizer (in a more intuitive form) penalizes the *model's Jacobian with respect to parameters*:
$$\mathcal{H} \equiv \left\Vert \frac{\partial T}{\partial A} \right\Vert^2_F + \left\Vert \frac{\partial T}{\partial B} \right\Vert^2_F + \left\Vert \frac{\partial T}{\partial C} \right\Vert^2_F$$
which can be viewed as a dual to L2 regularization: $\left \Vert  A \right \Vert^2_F   +  \left \Vert B \right \Vert^2_F   +  \left \Vert C \right \Vert^2_F$. This regularizer encourages the factors to learn full-rank, unitary embeddings. This is in contrast to L2 regularization which promotes low-rank solutions. This bias leverages the **Unitarity Theorem** of representation theory, which guarantees that every finite-dimensional representation is equivalent to a unitary representation. Therefore, by encouraging the model to consider only unitary matrix embeddings, we significantly **reduce the search space of solutions** without loss of generality.

### Appendix F: Alternative tensor factorizations
We provide comparison to common tensor factorization architectures and discuss their limitations.

* **Tucker decomposition** employs a core tensor $M$ and matrix factors: $T_{abc} = \frac 1 {n} \sum_{i,j,k} M_{ijk} A_{ai} B_{bj} C_{ck}$. While flexible, this suffers from a critical limitation:

  >learning the algebraic relationships between (a,b,c) of $T$ requires learning the relationships between (i,j,k) of $M$.

  This presents a recursive challenge, since  the matrix factors simply map individual *external* indices to individual *internal* indices (e.g. $A$ maps $a$ to $i$) without inherently simplifying the core learning problem. Consequently, Tucker decomposition severely overfits the training data and fails to generalize (Figure 12).

* **CP decomposition** utilizes only matrix factors: $T_{abc} = \frac 1 {n} \sum_{k} A_{ak} B_{bk} C_{ck}$, which is equivalent to HyperCube with diagonal embeddings: $A_{aki} = A_{ak} \delta_{ki}$, $B_{bij} = B_{bi} \delta_{ij}$, $C_{cjk} = C_{cj} \delta_{jk}$. Thus, CP can only capture commutative (Abelian) groups (e.g modular addition), which admit diagonal representations, but lacks the capacity to capture more complex operations (Figure 12).

### Appendix D: Hyperparameter sensitivity
We include Hyperparameter sensitivity analysis over a wide range of  learning rate, regularization strength, and initialization scale. HyperCube is robust across these variations with little impact on final generalization accuracy, suggesting that it does not require fine-grained tuning. This contrasts with L2 regularization, which is prone to weights collapsing to zero under strong regularization or small initialization. HyperCube lacks this local minimum at zero due to the quartic regularizer loss function, contributing to its robustness and bias towards full-rank solutions.

### Appendix E/G: Scalability
As mentioned above, HyperCube utilizes *Unitarity Theorem* to vastly reduce its effective parameter space. Despite its $O(n^3)$ parameters, HyperCube's scalability can be effectively managed by exploiting this **reduced parameter space and high parallelizability**.

* **Efficient GPU Implementation**: By leveraging the efficient parallelization of *einsum* operations in PyTorch, we demonstrate that the *runtime complexity* of HyperCube on GPUs remains nearly constant with increasing n (Appendix E). This allows for efficient training and inference even for moderately large groups.

* **Band-diagonal HyperCube** (Appendix G): To further improve scalability, we introduce a model variant with band-diagonal matrix embeddings. This variant effectively captures group structures while significantly reducing the parameter count to $O(n^2)$, allowing for scaling to larger groups without compromising performance.

---

### Author Response · Authors · 2024-11-25

Dear Reviewers,

As the rebuttal period nears its end, we want to express our sincere gratitude for your thoughtful feedback. We have diligently addressed your concerns through a revised manuscript, additional experiments, and detailed responses.  We hope these revisions clarify our contributions and provide a clearer understanding of the proposed model and its motivations.

Please don't hesitate to reach out if you require further clarifications or have any remaining questions. We are happy to provide more detailed explanations or conduct new experiments as needed.

We appreciate your time and consideration and hope that you will take these new results and revisions into account when making your final assessment.

---

### Author Response · Authors · 2024-12-02

Dear Reviewers,

As the discussion phase concludes today, we kindly ask if you could confirm whether our response has adequately addressed the concerns you raised. If so, we would greatly appreciate it if you could reflect this in an updated review score.

Thank you for your time and consideration.

---

### Meta-Review · Area_Chair_P2Vz · 2024-12-24

**Metareview:**

This paper presents a method for discovering groups and their representations. The proposed method is differentiable and leverages the representation theory of finite groups. The differentiability aspect is especially appealing in this context because it allows group axioms to be integrated into deep learning frameworks.

The reviewers appreciated that the paper is addressing a  long-standing problem, i.e., integrating group theory with deep learning.

There were some concerns from Reviewer wdaY regarding comparison of HyperCube with regularizers other than L2, and also about computational and run-time complexity of the proposed method.

Reviewer SQSt had some concerns regarding narrow focus on SOC tasks and limited comparisons with baselines.

Reviewer 13RW expressed some concerns regarding scalability (O(n^3))

The authors provided detailed answers to some of these concerns but there still remained some disagreement on others (e.g., alternate regularizers). That being said, such outstanding issues are not super-critical in my opinion given the other strengths of the paper, and follow-up works can possibly take up some of these investigations.

In view of the above, I recommend the paper for acceptance.

**Additional Comments On Reviewer Discussion:**

There were some concerns from Reviewer wdaY regarding comparison of HyperCube with regularizers other than L2, and also about computational and run-time complexity of the proposed method.

Reviewer SQSt had some concerns regarding narrow focus on SOC tasks and limited comparisons with baselines.

Reviewer 13RW expressed some concerns regarding scalability (O(n^3))

The rebuttal provided detailed answers to some of these concerns.The post-rebuttal discussion on the paper was somewhat limited and there still remained some disagreement on issues such as the possibility of using alternate regularizers. That being said, such outstanding issues are not super-critical in my opinion given the other strengths of the paper, and follow-up works can possibly take up some of these investigations.

---

### Decision · Program_Chairs · 2025-01-22

Accept (Poster)